# The role of minority language bilingualism in spotting agreement attraction errors: Evidence from Italian varieties

**Camilla Masullo** [1]*, **Alba Casado**[2], **Evelina Leivada**[3,4]

**1** Department of English and German Studies, Universitat Rovira i Virgili, Tarragona, Spain, **2** Department of Experimental Psychology, Mind, Brain, and Behavior Research Center, University of Granada, Granada, Spain, **3** Department of Catalan Philology, Universitat Autònoma de Barcelona, Barcelona, Spain, **4** Institució Catalana de Recerca i Estudis Avançats (ICREA), Barcelona, Spain

* camilla.masullo@urv.cat

**Data Availability Statement:** The data are held in the following repository:https://osf.io/j4zqg/?view_only=e52f1e4facb9474984148cefac087b51.

## Abstract

Bilingual adaptations remain a subject of ongoing debate, with varying results reported across cognitive domains. A possible way to disentangle the apparent inconsistency of results is to focus on the domain of language processing, which is what the bilingual experience boils down to. This study delves into the role of the bilingual experience on the processing of agreement mismatches. Given the underrepresentation of minority bilingual speakers of non-standard varieties, we advance a unique comparative perspective that includes monolinguals, standard language bilinguals, and different groups of minority language bilinguals, taking advantage of the rich linguistic diversity of the Italian peninsula. This comparative approach can reveal the impact of various sociolinguistic aspects of the bilingual experience across different bilingual trajectories. We developed an auditory acceptability judgement task in Italian, featuring Subject-Verb agreement mismatches. Participants evaluated the stimuli on a 5-point Likert scale and reaction times were recorded. The results do not reveal significant differences between the speakers of standard languages: Italian monolinguals and Italian-Spanish bilinguals. Instead, significant differences are found between monolinguals and the two groups of minority language bidialectals, as well as between the bidialectal groups themselves: Italian-Pavese bidialectals were faster than both Italian-Agrigentino bidialectals and Italian monolinguals, while Italian-Agrigentino bidialectals were less accurate than both Italian-Pavese bidialectals and Italian monolinguals. This intricate picture is explained through variables associated with second language use and language switching. Our findings suggest that if bilingualism is viewed as a yes/no phenotype, it is unavoidable that the bilingual experience will remain a mystery linked to intensely debated results. If, however, one accepts that bilingual adaptations are shaped by the environmental ecology of each trajectory, variation across bilingual processing outcomes is unsurprising. Overall, we argue that specific sociolinguistic factors behind each bilingual experience can reveal where bilingual adaptations on language and cognition stem from.

**Funding:** This work was supported by the European Union's Horizon 2020 research and innovation programme under the Marie Skłodowska-Curie grant agreement n˚ 945413 and from the Universitat Rovira i Virgili (URV) through 1 Martí i Franquès COFUND Doctoral Fellowships to CM. EL acknowledges funding from the Spanish Ministry of Science and Innovation (MCIN/AEI/ 10.13039/501100011033) under the research project No. PID2021-124399NA-I00. The funders had no role in study design, data collection and analysis, decision to publish, or preparation of the manuscript.

**Competing interests:** The authors have declared that no competing interests exist.

# Introduction

Bilingualism has been associated with behavioral and anatomical effects stemming from the presence of more than one linguistic system in the brain [1–5 *inter alia*]. At both the behavioral and the anatomical front, such bilingual effects have been abundantly discussed, often in terms of positive, negative, and null findings that come from various linguistic and non-linguistic domains [6,7]. On the one hand, the constant need to deal with two linguistic systems [8–10] has been associated with better performance in executive functions (EFs), metalinguistic and metacognitive awareness, abstract reasoning, and problem-solving [11–14]. On the other hand, studies testing EFs [15–17], as well as linguistic domains such as semantic and letter fluency [18,19] have often reported negative or null effects, leading to an apparent inconsistency of results [20] and an ongoing debate about the nature of the observed bilingual adaptations [21–24].

Seeing that the domain of EFs has given rise to largely contestable results, a possibly more reliable glimpse into the effects of bilingualism on cognition could come from focusing on language. Since the bilinguals' cognitive effort concerns managing different languages, bilingual effects in language processing are expected [25], especially when the tested stimuli take advantage of the parser's limitations. Indeed, although our cognitive parser can compute complex linguistic constraints, it is also likely to fail in the processing of some structures [26]. The parser's (in)success in computing certain structures has been described in terms of selective fallibility to the so-called *grammatical illusions*, which refer to stimuli that trick us in such a way that an ungrammatical sentence is considered acceptable [26].

One type of grammatical illusion concerns sentences that feature agreement attraction errors [26]. Such errors occur when a linguistic element and its grammatical controller do not agree. This lack of agreement is caused by a disrupting "distractor", which lies between them (1): Instead of agreeing with its controller, the mismatching element is attracted by the nearby distractor and follows its agreement features [27].

(1) *The key to the cabinets are rusty [28].

While the resulting sentence is ungrammatical, users do not consistently recognize it as such, primarily because the parser still computes agreement, albeit inaccurately, on the wrong element (i.e., the distractor). Therefore, the term "illusion" is used to describe agreement mismatches, as they deceive the parser by featuring agreement, but in a non-target way. Different theories have been proposed to explain this phenomenon. On the one hand, representational accounts [29–34], and specifically percolation accounts [29–31], have ascribed agreement attraction errors to ambiguous representations of the subject of the sentence. The main idea is that the mismatching number features of a distracting noun phrase (NP) adjacent to the subject are transferred to the subject of the sentence. As a result, the number features of the subject, which are used to compute the agreement on the verb, are faulty and lead to an agreement mismatch. On the other hand, retrieval accounts have interpreted agreement mismatches as a failure of the memory retrieval system [35–39]. Under this view, agreement mismatches stem from the retrieval process and are due to an overload of the working memory (WM) system, which is part of the EF domain. Rather than ascribing agreement mismatches to faulty representations of the subject itself, retrieval accounts posit that agreement mismatches arise due to the selection of an incorrect element, namely the distracting NP instead of the subject NP, during the retrieval process in the agreement region. Some studies within the retrieval account have suggested a positive correlation between enhanced EF abilities and lower susceptibility to attraction effects [40,41].

Among the agreement attraction phenomena which are more prone to interference effects, Subject-Verb number agreement stands out [26]. Subject-Verb agreement attraction errors

have been amply investigated in both production and comprehension. Regarding production, several studies have analyzed the impact of different NP features on attraction errors such as the NP number, animacy and length [28 for English], the nature of the NP number information [42 for English], the impact of number mismatch between NP and subject [43 for Dutch], the semantic distributivity of the head noun [44 for Italian and English], the linear proximity of the NP to the verb [30 for English], and the semantic integration of the NP to the head noun [45 for English]. Regarding the notional distributivity of the NP, some studies have considered the impact of the morphological richness of languages [46 for Mexican and Dominican Spanish], revealing that the richer the language morphology is, the fewer notional effects on agreement occur. Taken together, the results highlighted some cross-linguistic trends, the most frequent being that plural NPs elicit more attraction errors than singular NPs. At the same time, similar patterns of Subject-Verb agreement processing have also been observed in comprehension studies [27,29,47,48 for English; 49 for French]. Once again, plural NPs elicited higher attraction effects compared to singular NPs [50 for English]. Furthermore, the grammaticality of the stimuli was found to affect agreement attraction, giving rise to the so-called *grammatical asymmetry* for which attraction "eases the reading of ungrammatical verbs" [51, p. 147 for Spanish].

What both production and comprehension studies suggest is that agreement attraction errors are highly selective to specific morphological and syntactic patterns. However, the cognitive mechanisms behind the parser's fallibility are still unclear [52]. Following the retrieval accounts [53], it has been hypothesized that enhanced inhibitory control could prevent the parser from selecting the wrong NP for agreement [52]. In this context, we expect to find an effect of bilingualism on the computation of such grammatical illusions: If bilingualism leads to cognitive adaptations involving EFs, inhibitory control, and WM [9,54,55], testing grammatical illusions should reveal potential differences between monolingual and bilingual language processing [56].

This prediction is based on previous literature. Leivada, Mitrofanova, and Westergaard [25], for example, focused on comparative illusions and found that bilinguals were better at detecting them compared to monolinguals, but they were also slower in providing an answer as to the well-formedness of the stimuli. Regarding Subject-Verb agreement attraction errors, Foote [57] found that attraction was modulated by proficiency (i.e., more proficient bilinguals showed fewer attraction effects). The roles of age of acquisition (AoA) and proficiency were examined by Sagarra and Rodriguez [58], who found that Spanish monolinguals and English-Spanish bilinguals showed similar sensitivity to agreement violations. In particular, the processing patterns of adjacent Subject-Verb agreement in terms of reading times, gaze duration, and accuracy were found to positively correlate with perceptual salience, defined as "the ability of a stimulus to stand out from the rest and to attract attention by virtue of physical characteristics" [58, p. 16], and with L1 and L2 patterns of use, rather than AoA or proficiency. Similar rates of attraction for monolinguals and bilinguals were also reported by Lago and Felser [59], who compared German monolinguals and Turkish-German heritage speakers.

Crucially, while language processing has been examined in bilingual speakers of various standard/official languages, very few studies have focused on bilingual populations that use minority, regional, or non-standard varieties. Leivada [60] compared monolingual speakers of standard Greek and bidialectal speakers of Standard and Cypriot Greek in the detection of comparative illusions and reported a better performance for bidialectals. Regarding Subject-Verb agreement attraction errors, to the best of our knowledge, only Veenstra, Antoniou, Katsos, and Kissine [61] compared bilingual and bidialectal speakers. The tested populations were monolingual Dutch-speaking children, bilingual Dutch- and French-speaking children, and bidialectal Dutch- and West Flemish-speaking children. The three language groups did not

show any difference in attraction error production, but a correlation between attraction rates, verbal WM, and inhibitory control was found in all groups: participants with higher WM skills exhibited lower attraction rates compared to participants with weaker inhibitory control, who made more attraction errors.

Overall, considering the scarcity of research on bidialectal language processing, our study aims to add to the investigation of this severely understudied domain, by examining the processing of Subject-Verb agreement mismatches in various bilingual and bidialectal populations, hence advancing a unique comparative perspective. Specifically, our research questions (RQ) are: (I) Is there a difference in how monolingual, bilingual, and bidialectal speakers detect Subject-Verb agreement mismatches? (II) Is there an effect of specific sociodemographic and sociolinguistic variables of the bilingual experience on processing grammatical illusions that feature morphological mismatches?

RQ I ascribes our research to the frame of bilingual language processing, adding a new tile, which concerns the inclusion of bidialectal speakers. We use the term "bidialectal" for speakers of a standard and a non-standard, minoritized language. To address RQ I, we will focus on the linguistic landscape of Italy, which is particularly rich in terms of linguistic diversity. Besides standard Italian, a high number of local dialects is spoken from the north to the south of Italy, and they present great variability in terms of both structural and sociolinguistic traits [62]. Regarding the use of the term "dialect", an important terminological clarification is due. Italian dialects are not regional varieties of standard Italian, but independent linguistic systems that evolved directly from Latin and present their own structural features [63–65]. The major difference between standard Italian and these local dialects concerns the social prestige that speakers ascribe to them and their context of use. Although dialects are languages, the term "language" is usually reserved for the official, standard variety (i.e., standard Italian), while the term "dialect" indicates a variety that can be variably used in various contexts, often exclusively in unofficial and informal settings. Importantly, every Italian bidialectal community presents its own features in terms of dialect use, with the latter exhibiting considerable differences between northern and southern Italian regions. A prevailing trend emerges in favor of heightened dialect use in the southern regions, where standard Italian and dialects are more intricately intertwined [66,67]. We will consider two different Italian bidialectal groups, one from the north of Italy (i.e., Italian-Pavese bidialectals) and one from the south (i.e., Italian-Agrigentino bidialectals). Selecting two bidialectal groups that belong to different sociolinguistic realities offers a valuable opportunity to unveil the role of specific factors of diverse bilingual experiences as well as to tap into potential differences between them in terms of language practices. Regarding the phenomenon under study, Pavese and Agrigentino are similar to standard Italian in that they both inflect the verb for number and person to agree with the subject. However, Pavese presents an additional morphological marker for Subject-Verb agreement, which consists of a subject clitic pronoun preceding the verb [68–70].

RQ II is motivated by a rich line of studies that stress the importance of considering social and sociolinguistic aspects of the bilingual experience while investigating the cognitive effects of bilingualism [71–74 *inter alia*]. Considering different bilingual profiles may be the key to disentangling the role of factors such as proficiency, degree of use, social prestige, and personal attitudes towards different languages. Although bilingual and bidialectal speakers share the practice of regulating two linguistic systems, their main difference concerns the relation between these systems and their language use practices, as in the case of the Italian bidialectal communities we test. Since the Adaptive Control Hypothesis was developed [1], the context of use became a pivotal factor in defining different bilingual phenotypes. The role of context of use was further stressed by Beatty-Martínez and Titone [75], who propose that bilingual cognitive control is modulated by the degree of entropy, namely "the relative balance or diversity of

language use and/or exposure within and across communicative contexts" (p. 4). Under this view, comparing bilingual and bidialectal speakers entails focusing on different ecological systems, where the relation between the two languages is strongly shaped by the sociolinguistic context and the prestige ascribed to the linguistic varieties [76]. In their review of the effects of bidialectalism and diglossia on cognition, Alrwaita, Houston-Price, and Pliatsikas [77] highlight the importance of considering the context of use: In their words, "if the contexts in which language varieties are used is key in explaining the lack of consistency in the bidialectal literature, the inconsistent results of studies involving bilingual speakers might benefit from similar consideration" (p. 18).

In this context, the present study advances a comparative perspective that involves different populations (monolingual, bilingual, and bidialectal), while tapping into an aspect of language processing that has the potential to reveal whether juggling more than one language sharpens the cognitive parser in a way that makes it less vulnerable to grammatical illusions. More importantly, by compiling a detailed sociolinguistic profile for the different groups of participants under study and their language practices, the critical question of *what makes bilinguals different* will be addressed and variables concerning language practices, such as language switching, will hold a primary position. In addition, sociodemographic variables that have been found to potentially impact language processing, such as gender [78] and age [79], will be taken into account as control factors.

Based on previous literature, we expect different findings regarding RQ I. While it is plausible to anticipate comparable attraction effects in both monolingual and bilingual/bidialectal participants [56–59], we can equally expect to find some differences in the rates bilingual and bidialectal individuals detect Subject-Verb agreement mismatches in comparison to monolinguals, due to the ongoing language monitoring involved in the bilingual experience [25,60]. Regarding RQ II, we predict that these differences may be modulated by factors related to language use practices [59], which have been reported to affect cognitive control [1,75]. For both our RQs, the crucial dependent factors are accuracy in detecting agreement mismatches and reaction times (RTs) in providing a response. Besides the effect of language group, which will be investigated in our first analysis (RQ I), the effect of factors related to bilingual language use, such as time using the languages and switching practices, will constitute the independent variables of our further analysis (RQ II).

## Methods

### Participants

All participants were neurotypical adults. They were capable of providing informed consent and they gave their written informed consent prior to their participation in the study, in compliance with the Declaration of Helsinki. Most participants were recruited through invitations posted on social media platforms, while others were recruited in person. The recruitment period extended from December 2022 to April 2023. All participants completed the experiment on an online platform (Gorilla). In most cases, a researcher was actively involved during the recruitment phase of the experiment to ensure that participants could successfully access the provided link to the test. Subsequently, participants conducted the experiment in autonomy. The Ethics Committee for Research into People, Society and the Environment (CEIPSA) at Universitat Rovira i Virgili reviewed and approved the study protocol (approval number: CEIPSA-2022-TD-0032).

The original sample involved 278 participants, but 170 participants were excluded according to the following criteria: (i) not completing the task (n = 152), (ii) presenting more than 20% errors in acceptability rates of fillers (n = 14), (iii) not presenting the proper linguistic

profile to be included into one of the tested language groups (n = 4). The last criterion (iii) was assessed through participants' self-reported background measures. In particular, the 152 participants who did not complete the task were excluded because they only filled one section of the experiment, namely the background questionnaire, without starting or, in some cases, completing the acceptability judgement task before the end of the data collection session. The 4 participants excluded based on criterion (iii) were removed from the monolingual group. Our criterion to classify participants as monolinguals was based on pre-defined measures of language use. Specifically, only those participants who chose "never" or "few times" on a 5-point scale (i.e., "never", "few times", "sometimes", "often", and "always") that asked them about speaking, reading, and writing in the dialect/other language were included in the monolingual group. The final sample includes 108 participants who are divided into 4 groups: Italian-speaking monolinguals (n = 27), Italian-Spanish bilinguals (n = 27), Italian-Pavese bidialectals (n = 26), and Italian-Agrigentino bidialectals (n = 28). Table 1 shows the participants' demographics. The bilingual group includes both bilingual speakers of Italian and Spanish and trilingual speakers of Italian, Spanish, and Catalan (22% of the bilingual sample). The Pavese bidialectal group includes speakers of standard Italian and the Pavese dialect, and the Agrigentino group consists of speakers of standard Italian and the Agrigentino dialect. All bilingual and bidialectal participants reported having a high level of proficiency in their languages. Monolingual participants reported some basic or intermediate knowledge of either English or another language. Few monolingual participants also reported having some knowledge of their local dialect, but it was generally limited to low degrees of proficiency, and they did not use the local variety actively. A clarification about our use of the term "monolingual" should be made. With the term "monolingual", we indicate those participants who primarily master and use one language (i.e., standard Italian), but can have been passively exposed to other languages (i.e., local varieties in most cases or English as a foreign language at school). We believe that instead of the notion of "pure monolingual", which stems from the perception of bilingualism as a dichotomous condition, we should embrace the idea that being bilingual is a gradient status. Given that nowadays very few individuals have encountered only one language in their life, a more accurate classification of participants may involve expressions such as "more or less bilingual". Therefore, we opt to use the term "monolingual" just for the sake of simplicity, while emphasizing that it should be understood as the lower end of the bilingualism spectrum.

A demographic and sociolinguistic profile for each participant is available in Table 1 (for more detailed information see: https://osf.io/j4zqg/?view_only=e52f1e4facb9474984148cefac087b51).

Self-rated proficiency is measured on a 5-point scale where 1 is the minimum value and 5 is the maximum value. For bilinguals and bidialectals, the percentage of language use of Italian, the percentage of language use of L2, the percentage of language switching, and the mean age of onset of the L2 (in years) are also reported.

**Task.** An auditory timed acceptability task was developed and run in Gorilla (gorilla.sc) to collect both acceptability judgements on a 5-point Likert scale and RTs. Our task involved 120 auditory stimuli, split into 40 test items, 60 grammatical fillers, and 20 ungrammatical fillers, aiming for a 2:1 ratio between fillers and test items and a 1:1 ratio between grammatical and ungrammatical stimuli, following the experimental design proposed by Stowe and Kaan [80, p. 52; 81–83]. The stimuli were specifically created for this study and constitute original material available at: https://osf.io/j4zqg/?view_only=e52f1e4facb9474984148cefac087b51. All the test items (n = 40) had the same syntactic structure: These were ungrammatical sentences with a Subject-Verb agreement mismatch and a plural NP serving as a disrupting distractor between the subject and the verb. In all the test items, the subject was notionally non-distributive and grammatically singular. We split the test items into two conditions: half (n = 20) were

**Table 1. Participants' demographics.**

| | Monolinguals | Bilinguals | Italian-Pavese bidialectals | Italian-Agrigentino bidialectals |
|---|---|---|---|---|
| **N** | 27(18F) | 27(20F) | 26(19F) | 28(14F) |
| **Age** | 26.7 (3.9 SD) | 37.5 (10.9 SD) | 46.3 (16.5 SD) | 34.1 (13.2 SD) |
| **Education** | Secondary 5 | Secondary 7 | Primary 1 | Secondary 11 |
| | Tertiary 22 | Tertiary 20 | Secondary 13 | Tertiary 17 |
| | | | Tertiary 12 | |
| **Self-rated proficiency in Italian (1–5)** | | 4.78/5 | 4.58/5 | 4.54/5 |
| **Self-rated proficiency in the L2 (1–5)** | | 4.67/5 | 3.38/5 | 4.46/5 |
| **Percentage of daily language use—Italian** | | 51.96% | 72.15% | 58.04% |
| **Percentage of daily language use—L2** | | 38.52% | 22.62% | 37.57% |
| **Percentage of language switching** | | 59.26% | 47.12% | 54.64% |
| **Mean age of L2 onset** | | 14 y.o. (11.33 SD) | 0 y.o. (0 SD) | 0 y.o. (0 SD) |

presented in a high linguistic register (examples 2a and 3a), while the other half (n = 20) were presented in a low linguistic register (examples 2b and 3b). Linguistic register is defined as a variety of language shaped by different situational settings [84]. We included register variation in our stimuli to observe its potential effect on language processing and its interaction with the users' linguistic background. The test items in the high-register and low-register conditions were matched for semantic content. Each register condition had 10 items with animate NP distractors (2a and 2b) and 10 items with inanimate NP distractors (2a and 2b). The inclusion of both animate and inanimate NP distractors was driven by findings from previous literature [28], which reported an effect of animacy in the attraction rates of Subject-Verb agreement mismatches. The role of register and animacy of the test items will be separately discussed in another paper. We expect to find an effect of register variation on the detection of Subject-Verb agreement mismatches, further modulated by the participants' linguistic background. In terms of animacy, we aim to replicate previous findings [28], with animate distractors eliciting stronger attraction effects compared to inanimate distractors.

(2a) *Il documento dei poliziotti locali sono estremamente in disordine.
   The document.NOUN.SG of the policemen.NOUN.PL local be.3PL extremely in mess
   "The document of the local policemen are very messy."

(2b) *Il foglio degli sbirri comunali sono in un bordello assurdo.
   The sheet.NOUN.SG of the cops.NOUN.PL local be.3PL in a mess crazy
   "The sheet of the local cops are in a crazy mess."

(3a) *L'alloggio per le vacanze estive prevedono un costo elevato.
   The accommodation.NOUN.SG for the holidays.NOUN.PL summer have.3PL a cost high
   "The accommodation for the summer holidays are really expensive."

(3b) *La casa per le ferie estive costano un occhio della testa.

The house.NOUN.SG for the holidays.NOUN.PL summer cost.3PL an eye of the head

"The house for the summer holidays cost an arm and a leg."

Fillers involved both grammatical and ungrammatical sentences. The grammatical fillers consisted of sentences with correct Subject-Verb agreement (n = 40), which presented the same syntactic structure as the test items, and sentences with a different structure from the test items, which involved correct auxiliary choices for the verbs (n = 20). The ungrammatical fillers included sentences with wrong auxiliary choices for the verbs (n = 20).

All the linguistic stimuli were presented in standard Italian. Before the experiment started, a brief warm-up session was run to ensure that participants had understood the task correctly and had set the audio of their devices properly. All participants encountered all the test items, which were presented in a randomized order. Participants listened to them one by one and were asked to judge the sentence on a 5-point Likert scale with the following values: 1 = "Completely wrong", 2 = "Wrong", 3 = "Neither wrong nor correct", 4 = "Correct", 5 = "Completely correct".

Participants did not have the option of skipping a sentence or listening to it twice. RTs were recorded as soon as the participant selected a value on the Likert scale. Then, the next auditory stimulus was automatically played. Before the experiment, all participants completed a detailed sociolinguistic questionnaire, which was based on the Language and Social Background Questionnaire (LSBQ) [85]. The entire experiment (i.e., background questionnaire and acceptability judgement task) lasted between 30 and 40 minutes. The task, the dataset, the sociolinguistic questionnaire, and the R script used to run the analyses are available at: https://osf.io/j4zqg/?view_only=e52f1e4facb9474984148cefac087b51.

# Results

## Analyses

Since all 108 participants encountered all test items, which consisted of 40 ungrammatical sentences with Subject-Verb agreement mismatches, 8640 data points were collected, 4320 for each measure (i.e., acceptability judgements and RTs). Data analyses included both accurate and inaccurate responses to the test items. The standard logarithm ($RT' = \log_{10}(RT)$) was applied to normalize RTs, and a 2.5 SD filter was used to detect outliers. Consequently, 67/4320 data points have been removed from the RT measures (1.55%), and the corresponding acceptability judgements were also excluded. The results include 8506 data points for both acceptability judgements of test items and their corresponding RTs. The inclusion of both accurate and inaccurate responses and their corresponding RTs was done to comprehensively observe participants' behavior regarding agreement attraction errors, which is the main purpose of this study. By including RTs of both accurate and inaccurate responses, we seek to highlight potential trends in the time needed to give (in)correct responses, which have been highlighted in previous literature [60,86]. In particular, recent research on the processing of Subject-Verb agreement mismatches showed that inaccurate judgements are associated with slower RTs compared to accurate judgements [87]. Furthermore, given that Italian-Pavese and Italian-Agrigentino bidialectals have never been examined in the processing of Subject-Verb agreement mismatches, and in general, in language processing research, we opted not to exclude a priori a significant portion of our database to entirely observe the processing behavior of these unstudied populations.

We used the lme4 package (version 1.1.33) in R [88] to run both a generalized linear mixed-effects model (GLME) and a linear mixed-effects model (LME) [89–94].

## Accuracy analysis

First, we explored how the judgements of our 4 language groups differ in terms of accuracy (RQ I). To this aim, we selected a sum contrast for the Group variable, such that the monolingual group was set as the baseline level. The acceptability judgements on a 5-point Likert scale were coded as 1 for Accurate (i.e., judgements corresponding to either 1 or 2) and 0 for Inaccurate (i.e., judgements corresponding to 3, 4, or 5).

The GLME included accuracy as the dependent variable. As fixed effects, we included language group ("Monolinguals", "Bilinguals", "Agrigentino", "Pavese"). As control factors, we included the animacy of the NP distractors (sum contrast, two levels = "Animate", "Inanimate"), the register of the items (sum contrast, two levels = "High register", "Low register"), the chronological age of the participants (scaled), and gender (sum contrast, two levels = "Male", "Female"). As random intercepts, we included participants and items. We first fitted the maximal model following Barr, Levy, Scheepers, and Tily's recommendation [95], and if there were no convergence or singularities, we simplified it following Barr, Levy, Scheepers, and Tily [95] and Bates, Mächler, Bolker, and Walker [90]. To be more specific, we started by removing the interactions in the slopes, then we proceeded to remove the slopes with lower explained variance until convergence was reached. The final model included both participants and test items as random intercepts.

To observe the potential role of variables related to bilingual and bidialectal language practices (RQ II), we built a further model that did not include monolingual participants. In this model, we set the sum contrast for the Group variable with the bilingual group as the baseline. We included accuracy as the dependent variable. We kept language group as a fixed factor ("Bilinguals", "Agrigentino", "Pavese"). As additional fixed factors, we added the percentage of language switching (scaled), the percentage of Italian language use (scaled), the percentage of second language use (scaled), and their interactions with language groups. The percentage of language switching was calculated considering the mean value between the frequency of switching that participants reported for different contexts (i.e., home, university/work, other places) and with different interlocutors (i.e., relatives, friends, strangers). We kept the same control factors as the basic model, namely animacy, register, age, and gender, and both participants and test items as random intercepts. In both models, we consider significant any fixed effect with a t-statistic value not included between -2 and 2.

## RTs analysis

Besides accuracy, we explored how the judgements of the different language groups differ in terms of RTs (RQ I). Moreover, we examined whether RTs were modulated by acceptability judgements. To this aim, we selected a sum contrast for the Group variable, such that the monolingual group was set as the baseline level. To shed light on the role of acceptability judgements on RTs, we transformed the 1–5 Likert score to get a normal distribution by scaling the continuous variable. The LME included log-transformed RTs as the dependent variable. As fixed effects, we included the language group of the participants ("Monolinguals", "Bilinguals", "Agrigentino", "Pavese"), the acceptability judgement given to the stimulus (scaled), and the interaction between them. As control factors, we included the animacy of the NP distractors (sum contrast, two levels = "Animate", "Inanimate"), the register of the items (sum contrast, two levels = "High register", "Low register"), the chronological age of the participants (scaled) and the biological sex of the participants (sum contrast, two levels = "Male", "Female"). As random intercepts, we included participants and items. As we did for accuracy, we first fit the maximal model and if there were no convergence or singularities, we simplified it [95]. Again, we started by removing the interactions in the slopes, then we removed the

slopes with lower explained variance until reaching convergence. The final model included both participants and test items as random intercepts.

To inquire about specific variables of the bilingual experience (RQ II), we did a further analysis including both language use and language switching for bilingual and bidialectal participants. Once again, we excluded monolingual participants and we set the sum contrast for the Group variable with the bilingual group as the baseline level. This second LME for RTs included log-transformed RTs as the dependent variable. As fixed factors, we kept the language group of the participants ("Bilinguals", "Agrigentino", "Pavese"), the acceptability judgement given to the stimulus (scaled), and the interaction between them. As additional fixed factors, we added the percentage of language switching (scaled), the percentage of Italian language use (scaled), the percentage of use of the other language (scaled), and their interactions with language groups. The control factors were the animacy of the NP distractors, the register of the test items, age, and gender, and both participants and test items were set as random intercepts.

### The effect of language group on Accuracy and RTs–RQ (I)

**Accuracy.** Fig 1 shows the mean accuracy rates for each language group. Italian-Agrigentino bidialectals record the lowest accuracy rates compared to all the other language groups. Italian-Pavese bidialectals show the highest rates, followed by monolinguals and bilinguals, who record similar accuracy values. Setting the monolingual group as the baseline level in our model, a significant difference in accuracy rates is found only between monolinguals and Italian-Agrigentino bidialectals (t = -3.46, S1 Table, Supporting Information), with bidialectals performing worse than monolinguals.

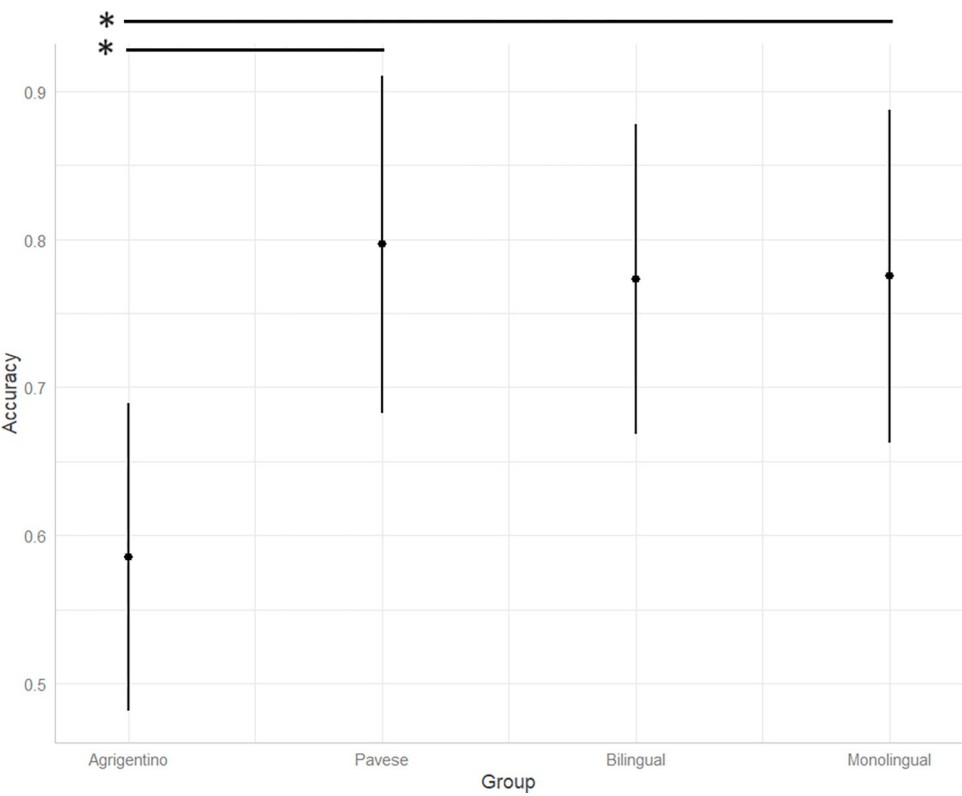

**Fig 1. Accuracy rates for each language group.** The x-axis represents the language groups, while the y-axis shows the mean accuracy rates from 0 (i.e., "Inaccurate") to 1 (i.e., "Accurate"). The vertical lines represent standard errors.

To inquire about possible differences between bilinguals and bidialectals, we reran the model setting the Italian-Pavese bidialectal group as the baseline level (S2 Table, Supporting Information). This analysis reveals a statistically significant difference between the two populations that use minority languages, with Italian-Agrigentino bidialectals recording lower accuracy rates compared to Italian-Pavese bidialectals (t = -3.46).

Regarding the control factors included in the model, a main effect of age is found (t = -3.05), with older participants being less accurate across all the language groups. Moreover, there is a main effect of Register with high-register sentences being evaluated less accurately compared to low-register ones (t = -2.06). To ensure that the model predictions are not influenced by the controlled factors, we calculated the collinearity coefficient between each fixed factor (VIF), which reveals that there is no correlation between them (S3 and S4 Tables, Supporting Information).

**RTs.** In our first RT analysis (S5 Table, Supporting Information), we find that Italian-Pavese speakers are the fastest to provide an answer, followed by the bilingual and the monolingual groups. The slowest group is the Italian-Agrigentino bidialectal group (Fig 2). Setting the monolingual group as the baseline level, the only statistically significant difference concerns the comparison between monolingual speakers and Italian-Pavese bidialectal speakers (t = -2.11). The only other comparison close to the significance threshold is the one between monolinguals and Italian-Agrigentino bidialectals (t = 1.80).

To further delve into the comparisons between the two groups of minority language bilinguals, we reran the analysis, setting the Italian-Agrigentino group as the baseline (S6 Table, Supporting Information). The new model reveals that the Italian-Agrigentino speakers are significantly slower than the Italian-Pavese group (t = -2.11).

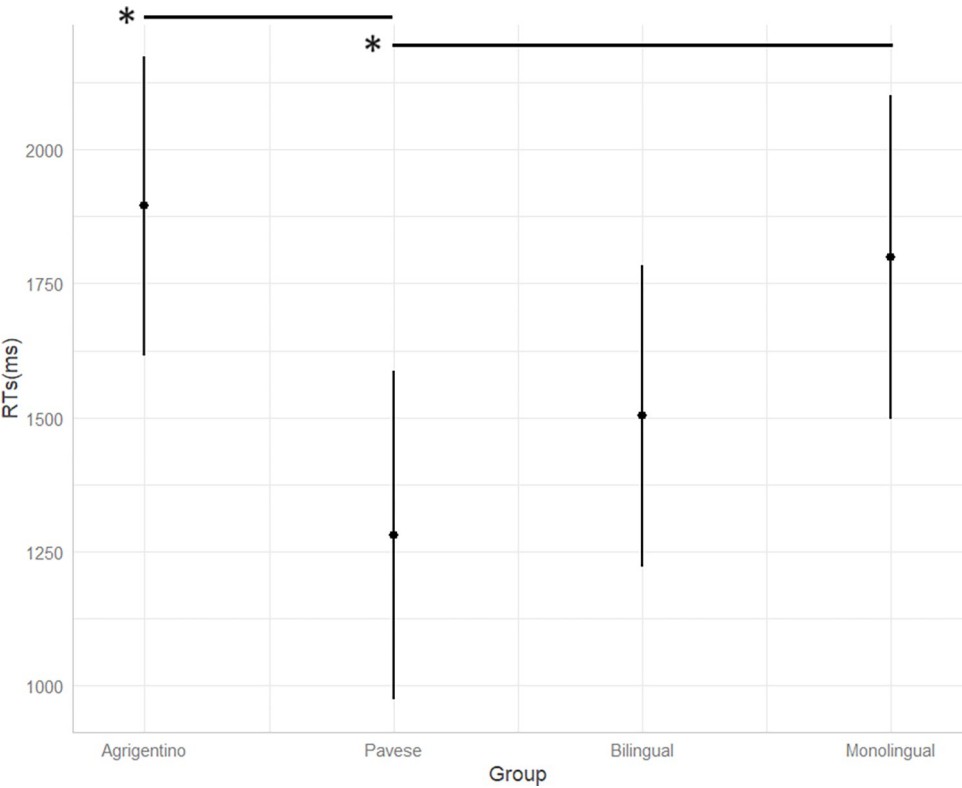

**Fig 2. RTs for each language group.** The x-axis represents the language groups, while the y-axis shows RTs in milliseconds recorded for each language group. The vertical lines represent standard errors.

 

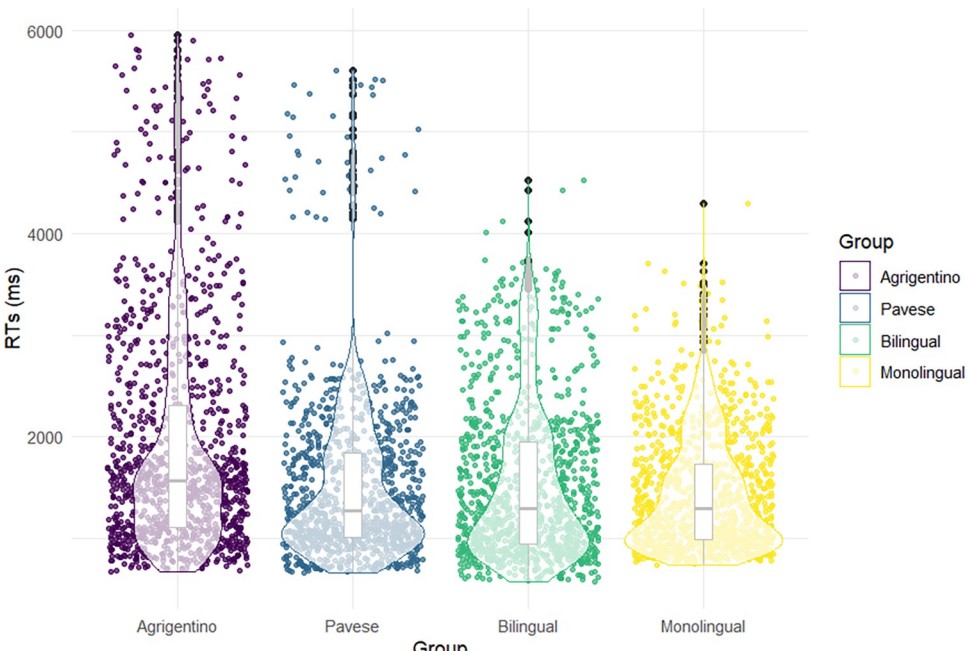

**Fig 3. Distribution of RTs for each language group.** The x-axis represents the distribution of RTs across various language groups. The y-axis reports RTs in milliseconds. The violin shapes represent data density, while the box plots represent standard deviations.

Fig 3 illustrates a more detailed distribution of RTs for each language group. Monolingual, bilingual, and Italian-Pavese bidialectal speakers show a similar distribution. Conversely, the Italian-Agrigentino bidialectal group shows the highest level of variation in the distribution of RTs.

We also find a main effect of the acceptability judgement (t = 9.39, S5 Table, Supporting Information). The results show that the accurate detection of the Subject-Verb agreement mismatches, which corresponds to lower acceptability judgements, is associated with reduced RTs, while the acceptance of incorrect stimuli, which is reflected in higher acceptability judgements, corresponds to longer RTs. In other words, the more participants think of the stimuli, the more likely it is that they provide an inaccurate answer, not spotting the morphological mismatch. It is worth recalling that in this analysis the judgement values are included as a continuous, scaled variable rather than a binary one, as was done for accuracy analyses. Interestingly, there is an interaction between acceptability judgements and language groups (t = -2.25, S5 Table, Supporting Information), revealing that while monolingual participants show a prominent difference between RTs associated with accurate vs. inaccurate judgements, such that inaccurate judgements are associated with longer RTs, the difference is less pronounced for Italian-Agrigentino bidialectal speakers, who exhibit only a minor difference in the time required to provide accurate or inaccurate judgements, as Fig 4 shows.

Consistent with what we reported for accuracy, a statistically significant effect of age is found (t = 3.76): older participants show slower RTs compared to younger participants across all language groups. To ensure that the model predictions are accurate, we calculated VIF, which reveals that there is no correlation between fixed effect factors (S7 and S8 Tables, Supporting Information).

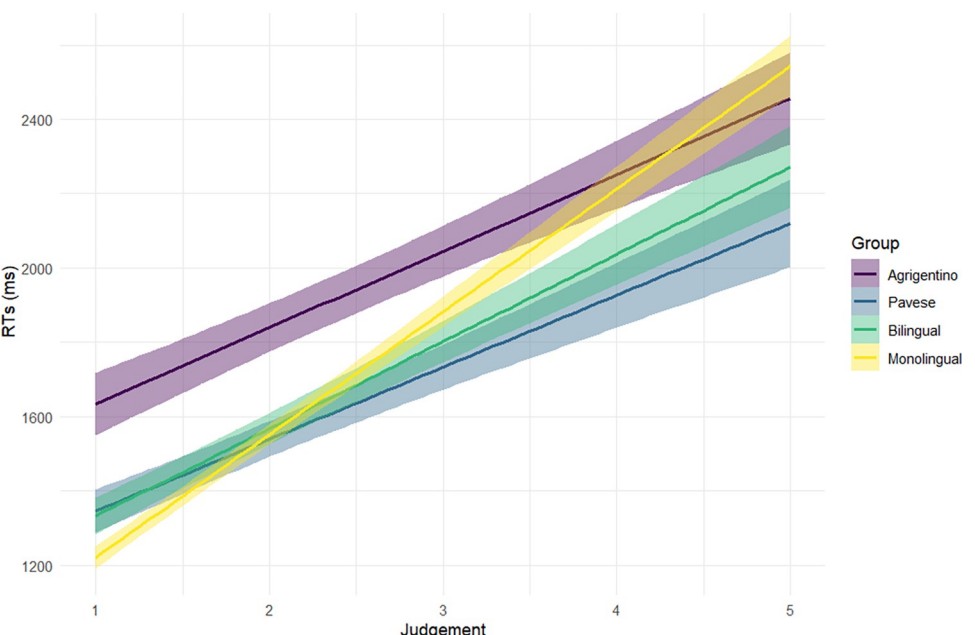

**Fig 4. Interaction between Acceptability Judgment and Language group on RTs.** The x-axis shows the acceptability judgements given to the stimuli. The y-axis reports RTs in milliseconds. The error ribbons represent 95% confidence intervals.

## The effect of language practices on Accuracy and RTs of bilingual and bidialectal participants–RQ (II)

**Accuracy.** Our second GLME with accuracy as the dependent variable was run to determine whether the percentage of use of Italian vs. the other majority or minority language, together with the frequency of language switching, modulates accuracy rates. For variables related to switching and percentage of language use, we do not find any statistically significant effect of language group on accuracy, thus we do not find differences between bilingual and bidialectal groups. Setting the bidialectal groups as the baseline instead of the bilingual group, the effect of language group is still not statistically significant.

In line with the previous model of accuracy, the results presented in the Supporting Information (S9 Table) show that there is a main effect of age and register. Older participants are less accurate than younger participants and low-register sentences record higher accuracy rates compared to high-register sentences.

**RTs.** For this second LME with RTs as dependent variable, we are interested in seeing whether variables related to the language practices of bilingual participants with minority vs. majority languages modulate RTs. As we did in the second GLME of accuracy, we inquire about the effect of the percentage of use of Italian vs. the other language (henceforth, L2), and the percentage of language switching on RTs. For this purpose, we set these variables as fixed factors, together with the language groups, setting the majority language bilingual group as the baseline. We find a main effect of the percentage of daily L2 use (S10 Table, Supporting Information). Fig 5 shows that higher percentages of daily use of L2 (i.e., dialect for bidialectals and Spanish for bilinguals) correspond to faster RTs in providing an answer. However, when examining the main effect of the percentage of use of Italian (i.e., participants' L1), we do not find any statistically significant effect on RTs.

There is a statistically significant interaction between language group and the percentage of language switching. In particular, bilinguals and Italian-Pavese bidialectals show opposite

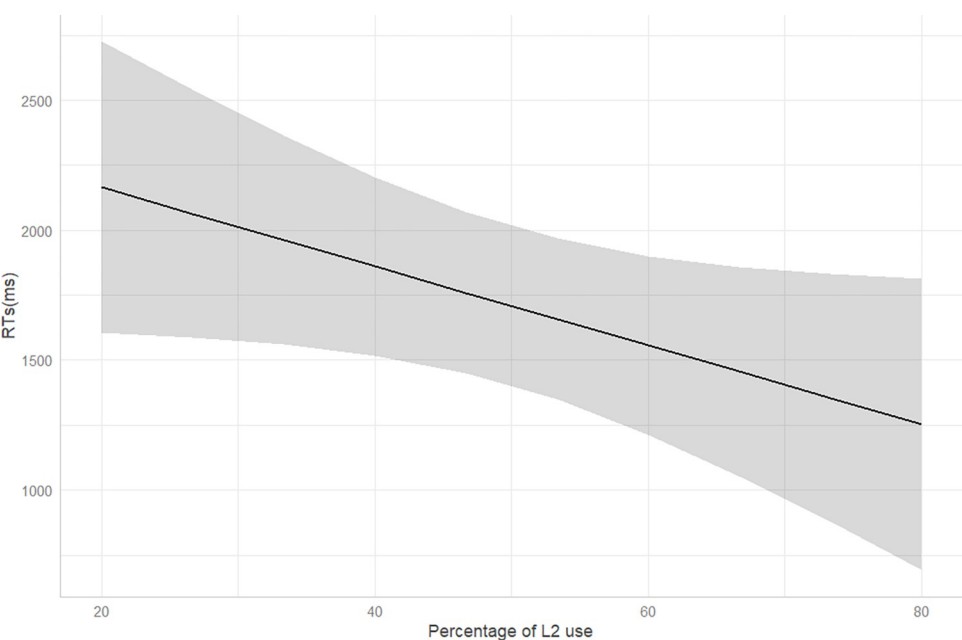

**Fig 5. Main effect of the percentage of L2 use on RTs.** The x-axis shows the percentage of time using the L2, while the y-axis reports RTs in milliseconds. The error ribbon represents 95% confidence interval.

trends (t = 2.33): Fig 6 shows that, for Italian-Pavese bidialectals shorter RTs correspond to lower percentages of language switching. For bilinguals, instead, higher percentages of language switching are associated with shorter RTs.

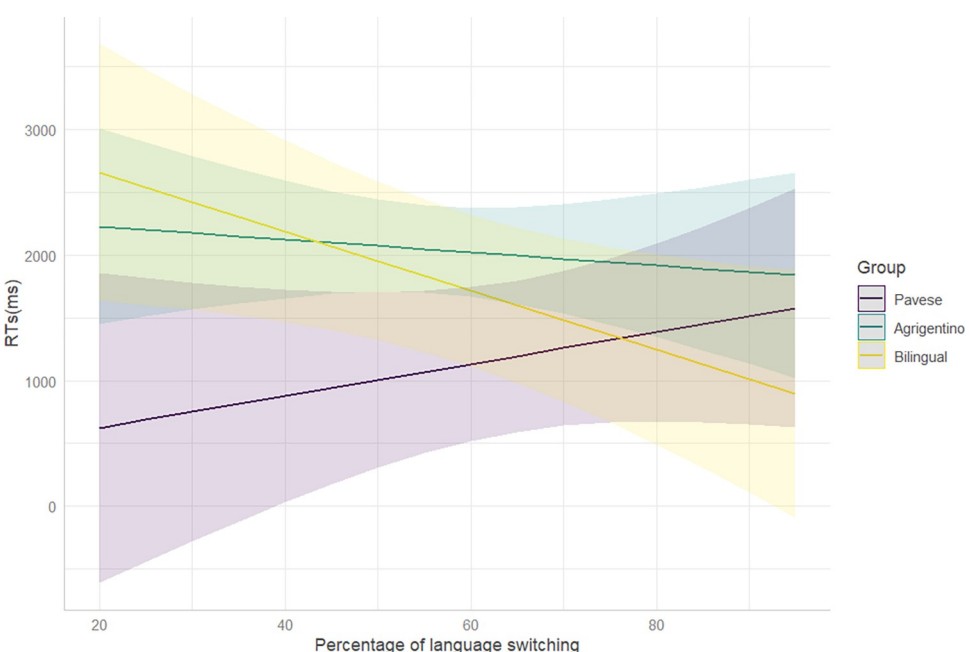

**Fig 6. Interaction between the percentage of language switching and Language group.** The x-axis shows the percentage of language switching. The y-axis reports RTs in milliseconds. The error ribbons represent 95% confidence intervals.

As in the previous model for RTs, a statistically significant effect of age on RTs is found (t = 3.21): older participants show longer RTs compared to younger participants.

In order to reveal potential differences between the two minority language-speaking groups, we reran the model setting the Italian-Agrigentino group as the baseline (S11 Table, Supporting Information). We find an interaction between language group and percentage of language switching. In particular, we observe a statistically significant difference between Italian-Agrigentino speakers and both Italian-Pavese speakers (t = 2.33) and bilingual speakers (t = -2.87). Contrary to what we found for bilinguals and Italian-Pavese speakers, the performance of the Italian-Agrigentino group does not show crucial changes in RTs according to higher or lower percentages of language switching.

## Discussion

The goal of the present study is to explore how bilingualism influences the linguistic processing of agreement attraction errors. The language of testing is Italian, which amounts to the only native language of the monolingual group and one of the native languages of the bilingual/bidialectal groups. In this regard, three possible scenarios have been proposed [56]: (i) greater processing difficulties for bilinguals compared to monolinguals, (ii) better bilingual performance in spotting agreement mismatches, since bilinguals' executive control components are regularly trained to suppress linguistic interference, and (iii) similar attraction effects between bilinguals and monolinguals. Our overall findings do not squarely fit into one of these predictions; instead, they fall into different scenarios according to the linguistic profile of the participants.

This study addresses 2 RQs: first, we are interested in observing whether there is a difference in monolingual and (minority vs. majority language) bilingual processing in comprehension tasks. Second, we want to shed light on whether there is an effect of specific sociodemographic and sociolinguistic variables of the bilingual experience, such as language switching practices and language use, on the processing of morphological mismatches. Regarding RQ I, the comparison between monolingual, bilingual, and bidialectal processing of attraction errors does not reveal significant differences between monolingual and bilingual speakers of standard languages (i.e., Italian and Spanish) either in accuracy or in RTs (Figs 1 and 2). This result is in line with previous literature, which reported similar processing outcomes for monolingual and bilingual speakers of standard languages [see 56 for Greek-German bilinguals; 58 for Spanish-English bilinguals; 59 for Russian-German bilinguals]. Additionally, a common trend is found for all monolingual, bilingual, and bidialectal participants: all participants show longer RTs when they give inaccurate, non-target answers. This finding too is in accordance with previous studies which reported faster decisions for correct judgements compared to incorrect judgements [58,96]. Since our experiment includes auditory stimuli that could only be played once, it is possible that sentences that sounded somewhat incorrect or dubious were carefully reinterpreted in an effort to search for and get a possible meaning assigned. Inevitably, this process increases the reaction window for making a decision and selecting a judgement.

Interestingly, a significant difference concerns monolinguals and minority language bilinguals: Italian-Agrigentino speakers show lower accuracy rates and longer RTs compared to their monolingual peers. Contrarily, Italian-Pavese bidialectals show higher accuracy rates compared to Italian-Agrigentino bidialectal participants and shorter RTs compared to both the monolingual baseline and the Italian-Agrigentino bidialectals. These findings stress the importance of differentiating between bilingual phenotypes when analyzing language processing outcomes [71,75,77]. Indeed, finding a significant difference between monolinguals and

bidialectals, but not between monolinguals and bilinguals of standard, majority languages leaves room for hypothesizing that variation in language processing is intimately connected to the sociolinguistic dimension of language development and use.

Our explanation about the lower accuracy of Italian-Agrigentino bidialectal speakers boils down to two factors. The first one concerns how Italian-Agrigentino bidialectals use their two languages. In this linguistic community, standard Italian and the Agrigentino dialect are not rigidly demarcated: the two linguistic systems coexist in different contexts and there is no strict norm about how and when one of the two languages should be used. The consequence is that Italian-Agrigentino speakers do not need to make a constant mental effort to keep their two linguistic systems strongly separated, so the amount of cognitive control employed in suppressing the interference of the second language is considerably reduced compared to other bilingual and bidialectal communities. The special role of dialects in the southern regions of Italy, and specifically in Sicily, where Agrigentino is spoken, is attested by data from Istituto Nazionale di Statistica (ISTAT) [66] which distinguishes southern regions from most of the northern Italian regions (with some exceptions such as Veneto and Friuli Venezia Giulia). In the specific case of Sicily, the prominent role of the dialect is also acknowledged in previous literature [97,98], which reports a strong fusion between the standard language (i.e., Italian) and the dialect in the Sicilian panorama. This picture is confirmed by the results of this study, where Italian-Agrigentino bidialectals report higher proficiency in their dialect and higher percentages of dialect use compared to their Italian-Pavese bidialectal peers. Moreover, the blurred boundaries between standard Italian and the Agrigentino dialect are confirmed by the short interactions that the experimenter had with some of the Italian-Agrigentino bidialectal participants during data collection: despite not being part of the Agrigentino ingroup, the experimenter was often addressed in a linguistic variety which presented evident dialectal elements and, sometimes, in the dialect itself. The sociolinguistic situation of Agrigento can be traced back to a dynamic of language cooperation [75]: driven by social conventions, bidialectal speakers in Agrigento might not feel the need to strongly monitor their linguistic behavior in terms of which is the appropriate variety to use. Importantly, having to continuously monitor external cues in order to be able to appropriately switch between languages engages cognitive control regions in the brain [99]. In terms of bilingual effects on language processing, the lack of constant exercise in inhibiting one of the two languages would explain the absence of an overall bilingual advantage in suppressing distracting information [52,53], such as the distractor behind the mismatching number in our stimuli. The wide degree of freedom with respect to language use and the low entropy that characterize language practices in Agrigento would not be comparable to a situation of language competition [75] nor to the dual language context described by the Adaptive Control Hypothesis, which instead could lead to enhanced inhibitory control [1].

Why would Italian-Agrigentino bidialectals perform worse than monolinguals in terms of spotting attraction errors (Fig 1), if monolinguals do not develop any special inhibition abilities either? This question leads us to our second point about the performance of Italian-Agrigentino bidialectals, which concerns the linguistic features of the varieties spoken in Agrigento. More specifically, the boundaries between the use of Italian and the dialect are very nuanced in Agrigento. The thick linguistic contact between Italian and Agrigentino results in a dense exchange of linguistic traits from one system to another [65,100]. Thus, the type of Italian spoken in Agrigento is strongly characterized by dialectal features, and the same happens to the dialect, which includes linguistic elements of Italian, especially in the lexical domain. The higher percentages of language switching reported by Italian-Agrigentino bidialectals (Table 1) further support this claim: in those sociolinguistic contexts where Italian and the dialect overlap across different communicative domains, as happens in Agrigento, the frequent

language-switching practices lead to a fusion between the codes, something less likely to happen in contexts where the two codes are kept more separated [101].

As a result, Agrigento is characterized by a linguistic *continuum* where standard Italian is deeply influenced by dialectal traits [see 102 for Sicily; 103–106 for Italy]. In this context, we explain the results of the Italian-Agrigentino group by highlighting the constant use of a strongly influenced linguistic system, where traits belonging to another language (in this case, the dialect) are not only accepted but may further lead to a higher tolerance with respect to what does not conform to the expected linguistic norm (e.g., grammatical deviations such as agreement mismatches). This is very likely to occur in sociolinguistic contexts where a standard language and a non-standard language co-exist since the latter is not defined by linguistic standardization [107–110]. In contrast, in bilingualism with standard languages, the two linguistic systems may be more rigidly demarcated, and rather than being on a linguistic continuum where structural traits from different languages are mixed, they are separately used by the speaker in a more defined code-switching mode [see 111 for the specific situation of Italian dialects; 112,113]. This brings to the fore the critical issue of *language proximity*: The closer two varieties are, the more likely it is that, if sociolinguistic conditions permit, they may result in a mixed lect that incorporates elements from both in certain contexts.

The concept of linguistic continuum introduced for the Italian-Agrigentino community can also explain our results for the Italian-Pavese bidialectals, who showed reduced RTs compared to both their Italian-speaking monolingual peers and the Italian-Agrigentino group. In this case, the common denominator behind the performance of Italian-Agrigentino and Italian-Pavese bidialectals concerns the relation between the dialect and standard Italian. Similar to what was described for Agrigentino, the boundaries between Pavese and Italian are more blurred compared to what would be expected if two standard languages were involved. However, what differentiates Pavia from Agrigento is the sociolinguistic function ascribed to the dialect: while in Agrigento the dialect and standard Italian co-exist in most communicative settings, the use of dialect in Pavia is limited to specific contexts, and the free interchange between dialect and standard Italian is less frequent [66]. This leads to a situation in which Italian-Pavese bidialectal speakers pay attention to the communicative context in which they use their dialect. Consequently, Italian-Pavese bidialectals need to (i) regulate the use of the dialect in settings where free switching is less common, and (ii) differentiate between two linguistic systems that exist along a continuum. The need of selecting the proper language to use and of disentangling between two tightly connected varieties could potentially strengthen their language control skills. This could explain the Italian-Pavese bidialectals' faster performance compared to monolinguals and Italian-Agrigentino bidialectals and their higher accuracy rates compared to Italian-Agrigentino bidialectals. Bilingual speakers who are used to making an effort to keep their two close systems separate could benefit from this training in a task that requires the inhibition of distracting information. Thus, the higher accuracy of the Italian-Pavese group could be interpreted as the effect of the specific sociolinguistic landscape in Pavia, where a more careful distinction between standard Italian and dialect is required.

Our interpretation of the results highlights once again the importance of considering the sociolinguistic dimension of the bilingual experience. Indeed, although the Italian-Agrigentino and Italian-Pavese groups both include bidialectal speakers of a majority and a minority language, they differ in terms of language practices. For the sake of clarity, the distinction made by Berruto [104] between social bilingualism, diglossia, dilalia, and bidialectalism (Table 2) could aid in understanding the degrees of variation which characterize our bidialectal communities. While the sociolinguistic context of Agrigento can be identified as a situation of dilalia that resembles bidialectalism, the Italian-Pavese community can be more accurately described as a situation of dilalia which, to some extent, is closer to diglossia. While in a situation of

**Table 2. Criteria for the identification of four linguistic repertoires.** Table adapted from Berruto [104, p206].

| Criteria | Social Bilingualism | Diglossia | Dilalia | Bidialectalism |
|---|---|---|---|---|
| Different sensibility between A and B | / | + | + | - |
| Use of both A and B in ordinary conversations | + | - | + | - |
| Clear functional difference between A and B | - | + | + | ? |
| Overlap of A and B in different domains | + | - | + | + |
| Standardization of B | / | + | - | - |
| B socially marked | / | - | + | + |
| Continuum between A and B | / | - | + | + |
| A has high social prestige | / | +/- | + | + |
| A and B both present in primary socialization | / | - | + | + |
| Possibility of promoting B as alternative code of A | / | + | + | - |
| Frequent code-switching between A and B | + | - | + | ? |
| Literary tradition for B | / | +/- | + | - |

diglossia, users associate each of their codes to specific social contexts (i.e., "high code"/standard Italian in official and formal settings vs. "low code"/dialect in informal settings), in a dilalic context, the two codes can overlap in different communicative situations [114–117].

The absence of different sensibility towards the two varieties which characterizes bidialectalism (Table 2) reflects the linguistic dynamics in Agrigento, where Italian-Agrigentino bidialectals might not need to strongly monitor switching between their varieties and, consequently, they treat them similarly. On the other hand, the lack of overlap between the two linguistic systems, which Berruto [104] ascribes to a situation of diglossia, is more typical of the sociolinguistic contexts of Pavia, where speakers tend to pay more attention to the proper language to use in different contexts.

With respect to potential differences between bilingualism with standard/majority vs. non-standard/minority languages, our analysis of the role of specific factors related to language use reveals interesting findings. The analyzed factors concern language use in terms of time speaking the languages and the percentage of language switching. Based on the comprehensive sociolinguistic questionnaire data we collected, our findings demonstrate that language processing outcomes can significantly change together with variables associated with specific language practices (i.e., RQ II). In particular, we find a negative relation between RTs and the percentage of L2 use. A possible explanation for this could be traced back to heightened awareness of the demarcation between the two distinct language systems, resulting from more time spent using the second language. This might lead to shorter RTs in a task involving just one of the two languages [118].

Besides the percentage of L2 use, another sociolinguistic factor that seems to play a role on RTs is language switching. An interesting difference is observed between bilinguals and Italian-Pavese bidialectals. While higher percentages of switching are associated with shorter RTs in bilinguals, the opposite trend is recorded for Italian-Pavese bidialectals (Fig 6). The negative relation between switching and response latencies found in bilinguals may suggest that the constant juggling between two languages trains the parser, leading to a faster performance [8–10,119]. However, the absence of a main effect of switching, the opposite patterns found in Italian-Spanish bilinguals and Italian-Pavese bidialectals, and the lack of interaction between switching and language group in the further analysis of bilinguals' and bidialectals' accuracy rates do not allow further speculations on the possible advantages of language switching for our bilingual participants. The main explanation could be related to the employed task. Seeing that monolinguals perform almost at ceiling in spotting attraction errors, task granularity

concerns [7] become relevant: if monolinguals already perform at ceiling in a task, possible bilingual effects will not be found, not because they do not exist, but because the task is not sensitive enough to reveal potential differences between the different groups.

Contrary to majority language bilinguals, Italian-Pavese bidialectal speakers show longer RTs when higher percentages of switching between standard Italian and dialect are reported (Fig 6). Once again, this finding can be attributed to the difference between the linguistic systems involved in the bilingual experience. Majority language bilinguals might have clearer boundaries between their two standard languages, while minority language bilinguals who use non-standard varieties may encounter greater challenges in distinguishing between two linguistic systems that exist on a continuum. This difficulty in disentangling the linguistic systems can potentially result in longer RTs during language processing for those bidialectal participants who report frequently switching practices between standard Italian and dialect, reflecting a greater degree of "fusion" between the two languages. Consequently, when the bidialectal participants are asked to make a judgement in just one of their linguistic systems, they may require additional time to disentangle their tightly interconnected languages and focus on only one of them. This interpretation of results might seem contradictory to our previous explanation for shorter RTs of Italian-Pavese bidialectals compared to their monolingual peers (Fig 2). However, there are two main differences between the two sets of results: (i) first, longer RTs are associated with higher percentages of language switching, a variable that was missing in the first analysis; (ii) second, RTs of Italian-Pavese bidialectals are longer compared to bilinguals, not to monolinguals. The crucial difference between the two apparently inconsistent findings should be ascribed to the role of language switching and how it is shaped by the bilingual communities under study. Language switching might be different for bilingual speakers of standard and non-standard languages. Despite the fact that we define both practices with the same term "language switching", bidialectals who report frequently switching between Italian and dialect might behave differently from bilinguals: rather than a proper switching between the standard and non-standard variety, a higher frequency of alternation between the two close systems might result in the use of a mixed variety, which allows for the coexistence of traits from both languages [111].

Moreover, considering our hypothesis that longer RTs are associated with frequent switching practices in Italian-Pavese bidialectals, it is reasonable to expect a similar performance for our Italian-Agrigentino bidialectal group. However, Fig 6 shows that the impact of language switching on the RTs of Italian-Agrigentino speakers is less pronounced than their Italian-Pavese peers. This result stresses once again the presence of variation across bilingual and bidialectal trajectories, which can be traced back to differences in the sociolinguistic contexts and, in turn, in linguistic practices between Pavia and Agrigento.

Taken together, the effects of time of language use and language switching confirm the importance of considering different variables of the bilingual experience and interpreting results according to the specific sociolinguistic context behind each bilingual profile. While some sociodemographic factors play a similar role across different bilingual populations, as shown by finding longer RTs for older participants in all language groups, the role of other variables seems to vary according to the social context of bilingualism and the status of the languages themselves.

## Conclusions

The present study focused on the effect of majority vs. minority language bilingualism on the processing of agreement attraction errors. Our results did not reveal significant differences between monolingual and bilingual speakers of standard languages in terms of accuracy or

RTs. Instead, differences were found between Italian-speaking monolinguals and the two bidialectal groups that use standard Italian plus a minority language: Italian-Agrigentino bidialectals were less accurate than monolinguals in spotting agreement mismatches and they were also slower in providing an answer, while Italian-Pavese bidialectal speakers showed a faster performance in RTs compared to their monolingual peers. Additionally, different processing outcomes were observed for bilinguals and bidialectals when variables related to language use were considered: frequent switching practices led to shorter RTs in bilingual speakers of standard languages, while Italian-Pavese bidialectal participants showed the opposite trend. Interestingly, some degree of variation was also found between the two bidialectal groups: Italian-Pavese bidialectals were faster and more accurate than their Italian-Agrigentino peers.

These results suggest the importance of differentiating between specific bilingual profiles and considering the environmental ecology of bilingual communities. Indeed, being bilingual is not limited to having more than one linguistic system in the brain. Rather, the key focus lies in understanding how these languages coexist and how they are employed in different contexts, settings, and registers. Variables such as standardization, minority vs. majority status, language use, and language switching should not be perceived as isolated values since they interact with each other and are shaped by the environment in which bilingual speakers live. A clear example comes from the bidialectal participants tested in this study. Their unique bilingual profiles entail different relations between the two languages, which are shaped by the sociolinguistic norms of use ascribed to each language. Indeed, language practices can influence the degree of "fusion" observed among the two co-existing varieties, leading to different processing outcomes. Thus, our findings corroborate the need for considering the sociolinguistic ecologies of bilingual communities [71,74,120], especially in situations where non-standard, minority, or regional varieties are involved, because these further invest the bilingual profile with significant variation. Devoting attention to the specific factors behind each bilingual experience could help us figure out where the cognitive effects of bilingualism stem from; an insofar open question with significant repercussions for the overall ability of the field to explain the results in terms of a coherent theory [22].

Among the limitations of our work, we would like to highlight the issue of adequate sampling and representation of minority language users who come from multidialectal communities that show considerable variation. Future work on larger and more diverse samples could add to our claims as well as clarify the impact of individual differences among participants. Furthermore, replicating our research while using different language groups will provide further insights into the role of various sociolinguistic variables, helping us to pinpoint the key factors that affect language processing. All in all, if we manage to ascribe bilingual effects to specific environmental conditions, the apparent inconsistency of results in bilingualism research could possibly be justified and explained as variation caused by the distinct sociolinguistic factors that synthesize every linguistic experience.

## Supporting information

**S1 Table. Fixed and random effects from the GLME of Accuracy, with the monolingual group as the baseline.** Accuracy rates are set as the dependent variable, language groups (i.e., "monolingual", "bilingual", "Agrigentino", and "Pavese") are set as fixed factors, while animacy, register, gender, and age are set as control factors.
(PDF)

**S2 Table. Fixed and random effects from the second GLME of Accuracy, with the Italian-Pavese bidialectal group set as the baseline.** Accuracy rates are set as the dependent variable, language groups (i.e., "monolingual", "bilingual", "Agrigentino", "Pavese") are set as fixed

factors, while animacy, register, gender, and age were set as control factors.
(PDF)

**S3 Table. VIF for the first GLME of Accuracy (S1 Table), with the monolingual group as the baseline.**
(PDF)

**S4 Table. VIF for the first GLME of Accuracy (S2 Table), with the Italian-Pavese bidialectal group set as the baseline.**
(PDF)

**S5 Table. Fixed and random effects from the LME of log-transformed RTs, with the monolingual group as the baseline.** Log-transformed RTs are set as the dependent variable, language groups (i.e., "monolingual", "bilingual", "Agrigentino", "Pavese") and "Judgement" are set as fixed factors. Animacy, register, gender, and age are set as control factors.
(PDF)

**S6 Table. Fixed and random effects from the first LME of log-transformed RTs, with the Italian-Agrigentino bidialectal group set as the baseline.** Log-transformed RTs are set as the dependent variable, language groups (i.e., "monolingual", "bilingual", "Pavese", "Agrigentino") and "Judgement" are set as fixed factors. Animacy, register, gender, and age are set as control factors.
(PDF)

**S7 Table. VIF for the first LME of log-transformed RTs (S5 Table), with the monolingual group as the baseline.**
(PDF)

**S8 Table. VIF for the first LME of log-transformed RTs (S6 Table), with the Italian-Agrigentino bidialectal group as the baseline.**
(PDF)

**S9 Table. Fixed and random effects from the second GLME of Accuracy, with the Italian-Pavese bidialectal group as the baseline.** Accuracy rates are set as the dependent variable. Language group (i.e., "Agrigentino", "Pavese", "bilingual"), "% of use of Italian", "% of use of the L2", and "% of switching" are set as fixed factors in the model and their interactions are also reported. Animacy, register, gender, and age are set as control factors.
(PDF)

**S10 Table. Fixed and random effects from the second LME of RTs, with the bilingual group as the baseline.** Log-transformed RTs are set as the dependent variable. Language group (i.e., "Agrigentino", "Pavese", "bilingual"), "Judgement", "% of use of Italian", "% of use of the L2", and "% of switching" are set as fixed factors in the model and their interactions with RTs are also reported. Animacy, register, gender, and age are set as control factors.
(PDF)

**S11 Table. Fixed and random effects from the second LME of RTs, with the Italian-Agrigentino bidialectal group as the baseline.** Log-transformed RTs are set as the dependent variable. Language group (i.e., "Pavese", "bilingual", "Agrigentino"), "Judgement", "% of use of Italian", "% of use of the L2", and "% of switching" are set as fixed factors in the model and their interactions with RTs are also reported. Animacy, register, gender, and age are set as control factors.
(PDF)

## Author Contributions

**Conceptualization:** Camilla Masullo, Evelina Leivada.

**Data curation:** Camilla Masullo.

**Formal analysis:** Alba Casado.

**Investigation:** Camilla Masullo.

**Methodology:** Camilla Masullo.

**Supervision:** Evelina Leivada.

**Writing – original draft:** Camilla Masullo.

**Writing – review & editing:** Alba Casado, Evelina Leivada.

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
