## [Decision Letter · Decision Letter 0]

28 Sep 2023

PONE-D-23-23745The role of minority language bilingualism in spotting agreement attraction errors: Evidence from Italian varietiesPLOS ONE

Dear Dr. Masullo,

Thank you for submitting your manuscript to PLOS ONE.  Therefore, we invite you to submit a revised version of the manuscript that addresses the points raised during the review process. Thank you again for submitting your manuscript ID PONE-D-23-23745 entitled "The role of minority language bilingualism in spotting agreement attraction errors: Evidence from Italian varieties" to Plos One. After careful consideration, we feel that it has merit but does not fully meet PLOS ONE’s publication criteria as it currently stands. The comments of  three reviewers are included at the bottom of this letter. You will see that they provide some substantial feedback on how the paper can be improved; while there are a number of different items to address, I think these should be doable and will lead to a clearer and more analytically robust paper. It is important, as the reviewer 2 suggests, that you describe in more detail the analyses that were carried out in order to can assess the discussion of the results found.

Please submit your revised manuscript by Nov 12 2023 11:59PM. If you will need more time than this to complete your revisions, please reply to this message or contact the journal office at plosone@plos.org. Please include the following items when submitting your revised manuscript:A rebuttal letter that responds to each point raised by the academic editor and reviewer(s). You should upload this letter as a separate file labeled 'Response to Reviewers'.A marked-up copy of your manuscript that highlights changes made to the original version. You should upload this as a separate file labeled 'Revised Manuscript with Track Changes'.An unmarked version of your revised paper without tracked changes. You should upload this as a separate file labeled 'Manuscript'.

We look forward to receiving your revised manuscript.

Kind regards,

Montserrat Comesaña Vila

Academic Editor

PLOS ONE

Journal Requirements:

This work was supported by the European Union’s Horizon 2020 research and innovation programme under the Marie Skłodowska-Curie grant agreement n° 945413 and from the Universitat Rovira i Virgili (URV) through 1 Martí i Franquès COFUND Doctoral Fellowships to CM. EL acknowledges funding from the Spanish Ministry of Science and Innovation (MCIN/AEI/10.13039/501100011033) under the research project No. PID2021-124399NA-I00. The funders had no role in study design, data collection and analysis, decision to publish, or preparation of the manuscript.

Reviewers' comments:

Reviewer's Responses to Questions

**Comments to the Author**

1. Is the manuscript technically sound, and do the data support the conclusions?

Reviewer #1: Yes

Reviewer #2: Partly

Reviewer #3: Yes

2. Has the statistical analysis been performed appropriately and rigorously? 

Reviewer #1: Yes

Reviewer #2: I Don't Know

Reviewer #3: Yes

3. Have the authors made all data underlying the findings in their manuscript fully available?

Reviewer #1: Yes

Reviewer #2: Yes

Reviewer #3: Yes

4. Is the manuscript presented in an intelligible fashion and written in standard English?

Reviewer #1: Yes

Reviewer #2: Yes

Reviewer #3: Yes

5. Review Comments to the Author

Reviewer #1: I miss work by Acuña-Fariña and Lago on attraction in Spanish. Also work by Foote and Bock on Mexican and Dominican, two dialects of Spanish.

The term 'grammatical illusion" needs to be explained, particularly why attraction is a an illusion.

By line 145 I felt a strong need to have a much clearer notion of what counts as a language and as a dialect. The authors say that the two 'dialects' studied are not regional varieties of Italian but two independent linguistic systems. If so, how come that are not languages? This gets worse when by line 190 they add the notion of register to the mix.

A short precis of the grammar of the 'dialects' involved should be added, especially with agreement in mind. Do all linguistic systems tested have the same morphological richness, the same type of subject verb agreement? Has notional distributivity being controlled?

Reviewer #2: Summary

The authors adopt a comparative approach to study bilingual adaptations. Besides monolinguals (Italians) and standard language bilinguals (Italian-Spanish), their sample included speakers of minority, regional, and non-standard varieties (minority language bilinguals: Italian-Agrigentino and Italian-Pavese). All participants performed an auditory acceptability judgment task in Italian with subject-verb agreement matches and mismatches. Performance was similar for monolinguals and standard language bilinguals, while differences were found between the monolingual group and the minority language bilinguals, and between the two groups of minority language bilinguals. The authors explain these differences in terms of sociolinguistic and ecological factors (specific use of language in their linguistic communities).

Evaluation

This is an interesting study. Including minority language bilinguals is a very welcome approach in the study of bilingual adaptations (I suspect that this form of bilingualism is the most frequent in Italy and in other European countries). The adoption of an ecological/sociolinguistic approach seems also very useful in helping identify factors potentially tied to language use of the non-standard language that might be related to performance in the standard language.

I think the paper would benefit from clarifying some aspects of the study, which I outline below.

Major points

1. Introduction.

a. I recommend that representational accounts (p. 4, line 80) be introduced a bit more extensively. For example, what would be ambiguous in these representations (and which representations, exactly)? Also, it is not clear to me how they differ from the retrieval account: wouldn’t ambiguity tax working memory as well?

b. How were the two Italian dialects chosen? The authors aim to test the “effect of specific sociodemographic and sociolinguistic variables of the bilingual experience on processing grammatical illusions that feature morphological mismatches,” but these variables are not mentioned in the Introduction. Was their selection made with some specific variables in mind? If not, the authors should stress the exploratory nature of their work. In addition, caution should be applied in framing the research question and interpreting the results: the authors use the terms “effect” and “impact,” but this is a correlational study and other expressions would be more appropriate (relationship, association, etc.).

c. RQ1: the research question is underspecified. For example, the authors use an acceptability task including grammatical mismatches; many comprehension tasks focus on the semantic analysis of more complex texts and do not include grammatical mistakes. I would identify what is exactly measured (at least in terms of conceptual variables) and theoretically justified based on the discussed literature.

d. It would be useful to end the Introduction with specific predictions, based on the reviewed literature. Ideally, this specification would indicate the critical dependent variables for the study and provide a map for the Results section.

2. Method.

a. P. 7, lines 183-184: “Following the experimental design of Stowe and Kaan [73] …]. The reference seems to be a textbook on methods. Have previous studies adopted a similar design and stimuli? If so, how were the results analyzed? [please see my comment below concerning accuracy and RT]

b. P. 10, participants section: almost half of participants did not complete the task (I assume the acceptability task, not the entire session). Given that the experimenter was often present (line 238), what were the reasons for not finishing the task? Also, 4 participants were excluded based on self-reported measures. Can you describe the inclusionary criteria adopted for the study?

c. Were the minority language bilinguals immersed in the Pavese and Agrigentino dialects from birth? What about the standard bilingual participants? Was there a difference in terms of age of acquisition between these groups?

3. Results. I am not an expert in mixed models analyses, but I am unclear on various aspects of the statistical analyses; I also wonder whether the Results section could be simplified:

a. Statistical analyses

a. The authors write they have run glmer analyses, but the script online only includes lmer analyses. That means that accuracy data, reduced to binary data (accurate and inaccurate, 1 vs. 0), were not analyze with a logistic analysis (a typical method is to use the binomial family in glmer for accuracy data).

b. The authors write that, for all the analyses, the final model included both participants and test items as random slopes. However, the script online only shows intercepts for the two random effects. In addition, the procedure for dealing with convergence issues needs to be clarified. Bates et al. (ref 77) offer minimal suggestions; if the authors used an optimizer, which one did they use? If needed, a variety of papers address this convergence issues (e.g., Barr et al., 2013; Barr et al., 2013, in Frontiers, on corrections; Bates et al., 2015, and Matuschek et al., 2017, on parsimonious models; Brauer & Curtin, 2018).

c. I wonder why the authors ran two sets of analyses, one including monolinguals and the other excluding them. Wouldn’t run only the most inclusive analysis and then set up paired comparisons be acceptable (e.g., with emmeans)? If would also simplify the Results section. For example, on p. 17 (with reference to Table 3), I am not sure I understand the need to rerun analyses with a different group as baseline.

d. The authors write that contrasts were set to sum contrasts, which is what appears in the R script. But the results seem to be set up in terms of treatment (or dummy) coding, in which one group (e.g., monolinguals) is the baseline. Please clarify.

e. What were the theoretical reasons to include Gender and Age as a fixed factor? These factors were not discussed in the Introduction. Do we expect any differences between/among genders?

f. Accuracy and RT analyses had the same number of data points. Did the RT analysis also include incorrect responses and grammatically correct sentences? Did accuracy data also include responses to grammatically correct sentences? Please clarify what set of data are typically analyzed for this task and which sets were used in the study.

g. VIF values are not reported, so it is difficult to estimate the extent of collinearity issues.

b. Figures and tables

a. Accuracy analyses were carried on accurate and inaccurate responses (a combination of values on the Likert acceptability scale). Figure 2 is providing mean accuracies for the various groups; thus, I am not sure what Figure 1 adds. Please clarify or remove the figure.

b. Log RT are difficult to interpret; can raw RT be reported somewhere?

c. Instead of the included tables in text, I think a more useful (because more direct) way to report the result of the linear mixed models analyses would be to report important values (e.g., t and coefficients) in the text and add the output of the analyses (the models that were tested) as supplementary materials.

d. Relatedly, I am confused about “interactions” between fixed factors and dependent variables (e.g., Table 5). Expressions regarding interactions between fixed factors and dependent variables are frequent in the manuscript. I suggest replacing them, as the term “interaction” is typically reserved for fixed factors (i.e., when a fixed factor moderates the effect of another factor on the dependent variable).

c. To make the Results section easier to read, I suggest linking specific analyses to specific research questions and replacing subsections with more informative titles (instead of main analysis and further analysis). Relatedly, the specific research questions should indicate the critical dependent variable. As mentioned above, I am still unclear why some analyses (e.g., correlations) were carried out on overall RT instead of the RT on mismatched sentences. Also, the first sentence of the Discussion clarifies that the goal of the study was “to explore how bilingualism influences the linguistic processing of agreement attraction errors.” However, it seems that analyses were run also on correct responses. In some cases, it is unclear what type of RT were analyzed. For example, on p. 25 (line 507 on), in the Discussion, the authors write about “the comparison between monolingual, bilingual, and bidialectal processing of attraction errors…” referring to Fig. 2 and 3. But it is not clear that the RT in Fig. 3 only refers to attraction errors.

4. Discussion. Given the problems outlined above (i.e., uncertainty regarding the specific dependent variables and analyses), I find it difficult to evaluate the Discussion. My suggestion would be to revise and reorganize the discussion after more targeted analyses. But I would like to highlight the following. Lower accuracy in Italian-Agrigentino speakers compared to monolinguals is tied to two factors: lack of a rigid demarcation regarding the context in which the two languages are used (with consecutively less need to monitor for external cues and inhibit one of the two languages, p. 27), and the nature of the relationship between the two languages, whose boundaries seem less rigid than for the other groups of bilinguals included in the study (p. 28; I assume these two factors are not completely independent). The latter aspect would create more tolerance in evaluating grammatical deviations (in the first language); the authors write “This dynamic is very likely to affect sociolinguistic contexts where a standard language and a non-standard language co-exist since the last is not defined by linguistic standardization [88-91].” If Agrigentino is not rigidly demarcated, I wonder how to interpret the 54.64% of language switching reported by speakers of Italian and Agrigentino (a percentage that is numerically higher than the one for Italian-Pavese bilinguals, 47.12%). Also, is there evidence that the presence of a linguistic continuum causes more tolerance in evaluating certain types of grammatical mismatches in the standard language or is it a speculation?

Minor points

- P. 5: “…the processing of adjacent Subject-Verb agreement was found to positively correlate with perceptual salience and language use, rather than AoA or proficiency.” Please describe the results of the study more in detail. What is, in this context, perceptual salience? And is perceptual salience important here? Also, which language in language use? L1 or L2? Also, “the processing of adjacent Subject-Verb agreement” refers to accuracy or errors?

- P. 6, end of first paragraph: “The three language groups did not show any difference in attraction error production, but a positive correlation between attraction rates, verbal WM, and inhibitory control was found in all groups.” This description refers to ref. 59 (the correct order of authors is Veenstra, Antoniou, Katsos, & Kissine, based on the original publication). I think the description is imprecise, as lower attraction rates were found in participants with higher verbal WM skills (so, the correlation was negative). In addition, participants who scored higher on inhibition scales (that is, showed higher switch costs and higher interference, indicating lower inhibition control) experienced higher attraction errors.

- Based on the script, multiple packages were used to analyze the data. However, the packages are not referenced in the References section. It is standard practice to include, in that section, all the R packages used for the analysis, along with the software used for R (e.g., R Studio) and their version, as different package versions might yield different results. Please refer to Meteyards & Davis (2020) for best practices for linear mixed-effects models in psychological science (another good paper is Brown, 2021).

- A number of concepts are introduced but not defined, for example, register, dilalia, disglossia. Please define these terms when they first appear in the text.

- Please spell out acronyms before using them (e.g., LSBQ, ISTAT; p. 9 and 27, respectively).

- Grammatical illusions are introduced on p. 3; the first example is on p. 8, in the Method section. I would provide an example at the outset (pp. 3 or 4) for readers not familiar with this literature.

- Method: the participant section typically precedes the description of the task (materials and procedure). What was the reason to invert their order?

- Results: Some effects were manipulated and statistical significant, but not much emphasis is put on them (e.g., register). Are they theoretically relevant?

- P. 25, first sentence of the discussion: “The goal of the present study is to explore how bilingualism influences the linguistic processing of agreement attraction errors.” I think it would be useful to add something regarding the language in which the errors were measured (Italian, which was the native language but perhaps not the only native language).

- P. 26: “it is possible that sentences that looked somewhat incorrect or dubious …”: I think “looked” should be replaced by “sounded,” as the stimuli were presented auditorily.

- P. 27, line 541: I think the word “language” follows “standard.”

Reviewer #3: This work presents a very interesting study that highlights the importance of taking into account the bilinguals’ profile and their socio-linguistic context for the comparison between monolinguals and bilinguals in tasks that involve grammatical illusions/agreement errors. It is also an innovative work, as it introduces the specific comparison between bilinguals of "standard/official" languages and bilinguals who use minority languages or dialects as L2. The two research questions are relevant and refer to these two different levels: one more typical of experimental psycholinguistics and another broader socio-linguistic one. Being able to bring both levels together and find satisfactory answers is, without a doubt, an ambitious goal. It seems that the authors succeed, at least partially.

The methodology is adequate, the task is a grammatical acceptability task involving auditory stimuli, and the materials used are well designed and controlled. The analyses are also correct. However, it is somewhat surprising that the manipulation of animacy and linguistic register is not justified and that the results obtained in relation to these factors are not presented here. I would like the authors to comment on the reasons for this decision and, even briefly or in a footnote, to indicate why they have included them and what differences can be expected between the different groups of participants in relation to these variables.

Finally, without questioning the rigor of this study, I would like a final explicit reflection from the authors regarding its scope/limitations. Given the enormous variability across different sociolinguistic contexts, and even across individuals, do the authors estimate that the size of the groups is sufficient and that these results would be easily replicable in the same context or in a different one? One might think that the task of finding consistent effects in cognitive/linguistic processing across bilingual studies is infeasible. Or that we could fall into ad hoc explanations that justify any type of effect, no matter how surprising it may seem.

6. PLOS authors have the option to publish the peer review history of their article (what does this mean?). If published, this will include your full peer review and any attached files.

Reviewer #1: No

Reviewer #2: No

Reviewer #3: No

---

## [Author Response · Author response to Decision Letter 0]

10 Nov 2023

Response letter

We would like to thank the Editor, Prof. Montserrat Comesaña Vila, and the three Reviewers for their generous and very helpful feedback. In this letter, we address all the points raised by the Reviewers and link them to the related changes in the revised version of the manuscript. The newly added parts as well as the parts that were significantly revised appear in red to facilitate easy identification and are also reported in this response letter.

Comments by Reviewer 1

We would like to thank the Reviewer for the valuable feedback, which contributes to improving the quality of our work. In the following points, we will describe the changes we applied:

● I miss work by Acuña-Fariña and Lago on attraction in Spanish. Also work by Foote and Bock on Mexican and Dominican, two dialects of Spanish. 

We would like to thank the Reviewer for the suggested references. We added these works in the Introduction section, on p. 5, and we also updated our Reference list accordingly. 

P. 5: “Regarding the notional distributivity of the NP, some studies have also considered the impact of the morphological richness of languages [46 for Mexican and Dominican Spanish], revealing that the richer the language morphology is, the fewer notional effects on agreement occur.”

P. 5: “Furthermore, the grammaticality of the stimuli was found to affect agreement attraction, giving rise to the so-called grammatical asymmetry for which attraction “eases the reading of ungrammatical verbs” [51, p 147 for Spanish].”

● The term “grammatical illusion" needs to be explained, particularly why attraction is an illusion. 

We explained the term “grammatical illusion” more in detail, with a specific focus on attraction errors, on p. 4, together with an example sentence.

P. 4: “This lack of agreement is caused by a disrupting “distractor”, which lies between them (1): Instead of agreeing with its controller, the mismatching element is attracted by the nearby distractor and follows its agreement features [27].

(1) *The key to the cabinets are rusty. [28]

While the resulting sentence is ungrammatical, users do not consistently recognize it as such, primarily because the parser still computes agreement, albeit incorrectly, on the wrong element (i.e., the distractor). Therefore, the term “illusion” is aptly used to describe agreement mismatches, as they deceive the parser by featuring agreement, but in a non-target way.”

● By line 145 I felt a strong need to have a much clearer notion of what counts as a language and as a dialect. The authors say that the two 'dialects' studied are not regional varieties of Italian but two independent linguistic systems. If so, how come that are not languages? This gets worse when by line 190 they add the notion of register to the mix. 

Essentially, they are languages. This comment has been addressed on p. 7 of the Introduction. We explain the use of the terms “language” and “dialect” in the Italian panorama.

P. 7: “The major difference between standard Italian and dialects concerns the social prestige that speakers ascribe to them and their context of use. Although dialects are languages, the term “language” is usually reserved for the official, standard variety (i.e., standard Italian), while the term “dialect” indicates a variety that can be variably used in various contexts, often exclusively in unofficial and informal settings. Importantly, every Italian bidialectal community presents its features in terms of dialect use, which exhibits considerable differences between northern and southern Italian regions. A prevailing trend emerges in favor of heightened dialect use in the southern regions, where standard Italian and dialects are more intricately intertwined [66,67].”

● A short precis of the grammar of the 'dialects' involved should be added, especially with agreement in mind. Do all linguistic systems tested have the same morphological richness, the same type of subject-verb agreement?

We added this information on pp. 7-8.

Pp. 7-8: “Regarding the phenomenon under study, Pavese and Agrigentino are similar to standard Italian in that they both inflect the verb for number and person to agree with the subject. However, Pavese presents an additional morphological marker for Subject-Verb agreement, which consists of a subject clitic pronoun preceding the verb [68-70].”

● Has notional distributivity being controlled? 

In the revised version, we specified this on p. 12.

P. 12: “In all the test items, the subject was notionally non-distributive and grammatically singular.”

Comments by Reviewer 2

● Summary: The authors adopt a comparative approach to study bilingual adaptations. Besides monolinguals (Italians) and standard language bilinguals (Italian-Spanish), their sample included speakers of minority, regional, and non-standard varieties (minority language bilinguals: Italian-Agrigentino and Italian-Pavese). All participants performed an auditory acceptability judgment task in Italian with subject-verb agreement matches and mismatches. Performance was similar for monolinguals and standard language bilinguals, while differences were found between the monolingual group and the minority language bilinguals, and between the two groups of minority language bilinguals. The authors explain these differences in terms of sociolinguistic and ecological factors (specific use of language in their linguistic communities).

Evaluation: This is an interesting study. Including minority language bilinguals is a very welcome approach in the study of bilingual adaptations (I suspect that this form of bilingualism is the most frequent in Italy and in other European countries). The adoption of an ecological/sociolinguistic approach seems also very useful in helping identify factors potentially tied to language use of the non-standard language that might be related to performance in the standard language.

I think the paper would benefit from clarifying some aspects of the study, which I outline below.

We would like to express our gratitude to the Reviewer for the positive assessment of our work and the very helpful comments. In the following points, we explain how we modified our paper according to the Reviewer’s suggestions.

Major points

● Introduction: I recommend that representational accounts (p. 4, line 80) be introduced a bit more extensively. For example, what would be ambiguous in these representations (and which representations, exactly)? Also, it is not clear to me how they differ from the retrieval account: wouldn’t ambiguity tax working memory as well? 

We specified this on p. 4. While number feature ambiguity is central to both accounts, representational accounts do not primarily address working memory costs. Instead, working memory is invoked within retrieval accounts to explain the process of choosing an incorrect element for agreement.

P. 4: “On the one hand, representational accounts [29-34], and specifically percolation accounts [29-31], have ascribed agreement attraction errors to ambiguous representations of the subject of the sentence. The main idea is that the mismatching number features of a distracting noun phrase (NP) adjacent to the subject are transferred to the subject of the sentence. As a result, the number features of the subject, which are used to compute the agreement on the verb, are faulty and lead to an agreement mismatch… Rather than ascribing agreement mismatches to faulty representations of the subject itself, retrieval accounts posit that agreement mismatches arise due to the selection of an incorrect element, namely the distracting NP instead of the subject NP, during the retrieval process in the agreement region.”

● How were the two Italian dialects chosen? The authors aim to test the “effect of specific sociodemographic and sociolinguistic variables of the bilingual experience on processing grammatical illusions that feature morphological mismatches,” but these variables are not mentioned in the Introduction. Was their selection made with some specific variables in mind? If not, the authors should stress the exploratory nature of their work. 

In the revised manuscript, our justification for selecting these two varieties is given on p. 7.

P. 7: “Importantly, every Italian bidialectal community presents its features in terms of dialect use, which exhibits considerable differences between northern and southern Italian regions. A prevailing trend emerges in favor of heightened dialect use in the southern regions, where standard Italian and dialects are more intricately intertwined [66,67]. We will consider two different Italian bidialectal groups, one from the north of Italy (i.e., Italian-Pavese bidialectals) and one from the south (i.e., Italian-Agrigentino bidialectals). Selecting two bidialectal groups belonging to different sociolinguistic realities offers a valuable opportunity to unveil the role of specific factors of diverse bilingual experiences and to tap into potential differences between them in terms of language practices.”

● In addition, caution should be applied in framing the research question and interpreting the results: the authors use the terms “effect” and “impact,” but this is a correlational study and other expressions would be more appropriate (relationship, association, etc.). 

We apologise for the confusion created by the use of inaccurate terminology in the previous version of the manuscript. This is not a correlation study. For the analyses, we used regression models, which reveal whether and in what way the independent variables (i.e., fixed effects) have an impact on the dependent ones (accuracy of the judgments and response time). Differently from correlation analyses, which assess the strength of the relationship between two variables, regression analyses uncover causal relationships between the variables, enabling us to discern the effects they exert on one another. Hence our choice to use terms such as “effect” and “impact”.

● RQ1: the research question is underspecified. For example, the authors use an acceptability task including grammatical mismatches; many comprehension tasks focus on the semantic analysis of more complex texts and do not include grammatical mistakes. I would identify what is exactly measured (at least in terms of conceptual variables) and theoretically justified based on the discussed literature. It would be useful to end the Introduction with specific predictions, based on the reviewed literature. Ideally, this specification would indicate the critical dependent variables for the study and provide a map for the Results section. 

Following the Reviewer’s suggestion, we specified our RQ1 on p. 7 and the variables of our analyses on p. 9, and we offer predictions on p. 9.

P. 7: “Specifically, our research questions (RQ) are: (I) Is there a difference in how monolingual, bilingual, and bidialectal speakers detect Subject-Verb agreement mismatches?”

P. 9: “…the critical question of what makes bilinguals different will be addressed and variables concerning language practices, such as language switching, will hold a primary position. In addition, sociodemographic variables that have been found to potentially impact language processing, such as gender [83] and age [84], will be taken into account as control factors.”

P. 9: “Based on previous literature, we expect different findings regarding RQ (I). While it is plausible to anticipate comparable attraction effects in both monolingual and bilingual/bidialectal participants [57-60], we can equally expect to find some differences in the way bilingual and bidialectal individuals detect Subject-Verb agreement mismatches in comparison to monolinguals, due to the ongoing language monitoring involved in the bilingual experience [25,60]. Regarding RQ (II), we predict that these differences may be modulated by factors related to language use and practices [59], which have been reported to affect cognitive control [1,75]. For both our RQs, the crucial dependent factors are accuracy in detecting agreement mismatches and reaction times (RTs) in providing a response. Besides the effect of language group, which will be investigated in our first analysis (RQ I), the effect of factors related to bilingual language use, such as time using the languages and switching practices, will constitute the independent variables of our further analysis (RQ II).”

● Method: P. 7, lines 183-184: “Following the experimental design of Stowe and Kaan [73] …]. The reference seems to be a textbook on methods. Have previous studies adopted a similar design and stimuli? If so, how were the results analyzed? [please see my comment below concerning accuracy and RT] 

Yes, many previous studies have adopted a similar design. In the revised version, we have added references on p. 12.

P. 12: “following the experimental design proposed by Stowe and Kaan [80, p. 52; 81, 82, 83].”

● P. 10, participants section: almost half of participants did not complete the task (I assume the acceptability task, not the entire session). Given that the experimenter was often present (line 238), what were the reasons for not finishing the task? 

We addressed this point by adding a clearer explanation of the data collection procedure on pp. 9-10.

Pp. 9-10: “In most cases, a researcher was actively involved during the recruitment phase of the experiment to ensure that participants could successfully access the provided link to the test. Subsequently, participants conducted the experiment in autonomy.”

● Also, 4 participants were excluded based on self-reported measures. Can you describe the inclusionary criteria adopted for the study? 

We added the inclusion criteria on p. 10.

P. 10: “In particular, the 4 participants excluded based on criterion (iii) were removed from the monolingual group. Our criterion to classify participants as monolinguals was based on pre-defined measures of language use. Specifically, only those participants who chose “never” or “few times” on a 5-point scale (i.e., “never”, “few times”, “sometimes”, “often”, and “always”) that asked them about speaking, reading, and writing in the dialect/other language were included in the monolingual group.”

● Were the minority language bilinguals immersed in the Pavese and Agrigentino dialects from birth? What about the standard bilingual participants? Was there a difference in terms of age of acquisition between these groups?

In the revised version, we reported additional information about Age of Onset in Table 1:

Mean age of L2 onset 14 y.o. (11.33 SD) 0 y.o. (0 SD) 0 y.o. (0 SD)

Since the Age of Onset of bilinguals was different from bidialectals, we wanted to ensure that this difference did not influence the results of our models. Thus, we rerun the models presented in the Results section of our paper, under the subsections “The effect of language group on Accuracy and RTs – RQ(I)” and “The effect of language practices on Accuracy and RTs of bilingual and bidialectal participants – RQ(II)”. We did not find a significant effect of AoO on either accuracy or RTs. Here, we also report the VIF values of the models, which show that the Age of Onset variable did not compromise the main effect of our fixed factors in our models.

Factor GVIF Df GVIF^(1/(2*Df))

Group 3.120037 3 1.208815328

Judgement 1.075872 1 1.037242652

Animacy 1.00135 1 1.000674553

Register 1.002768 1 1.001383181

Gender 1.058278 1 1.028726393

Age 1.453769 1 1.205723375

Age of Onset 2.310785 1 1.520126797

Group:Judgement 1.072206 3 1.011687443

VIF for the LME of RTs run on the whole sample, with the inclusion of Age of Onset as control factor.

Factor GVIF Df GVIF^(1/(2*Df))

Group 10.37238 2 1.794608004

Judgement 1.09879 1 1.048231826

% of Switching 1.473925 1 1.214053286

% of Italian language use 1.948302 1 1.395815918

% of second language use 2.597821 1 1.611775732

Animacy 1.001082 1 1.000540871

Register 1.003481 1 1.001738971

Age of Onset 2.565954 1 1.601859523

Gender 1.452705 1 1.205282276

Age 1.346478 1 1.160378282

Group:Judgement 1.094789 2 1.022898719

Group:% of Switching 2.386208 2 1.24287391

Group:% of Italian language use 2.704352 2 1.2823772

Group:% of second language use 4.088533 2 1.421974749

VIF for the LME of RTs run on the bilingual and bidialectal sample, with the inclusion of Age of Onset as control factor.

Factor GVIF Df GVIF^(1/(2*Df))

Group 3.126807 3 1.209252

Animacy 1.000054 1 1.000027

Register 1.000065 1 1.000033

Gender 1.05814 1 1.028659

Age 1.474059 1 1.214108

Age of Onset 2.286663 1 1.512172

VIF for the GLME of Accuracy run on the whole sample, with the inclusion of Age of Onset as control factor.

Factor GVIF Df GVIF^(1/(2*Df))

Group 10.53346 2 1.801535

% of Switching 1.485405 1 1.218772

% of Italian language use 2.036926 1 1.427209

% of dialect language use 2.562015 1 1.60063

Animacy 1.000066 1 1.000033

Register 1.0001 1 1.00005

Gender 1.4576 1 1.207311

Age 1.383743 1 1.176326

Age of Onset 2.527439 1 1.589792

Group:% of Switching 2.46278 2 1.252727

Group:% of Italian language use 2.870259 2 1.301608

Group:% of second language use 4.018884 2 1.41588

VIF for the GLME of Accuracy run on the bilingual and bidialectal sample, with the inclusion of Age of Onset as control factor.

● Results: I am not an expert in mixed models analyses, but I am unclear on various aspects of the statistical analyses. I also wonder whether the Results section could be simplified.

Statistical analyses: The authors write they have run glmer analyses, but the script online only includes lmer analyses. That means that accuracy data, reduced to binary data (accurate and inaccurate, 1 vs. 0), were not analyzed with a logistic analysis (a typical method is to use the binomial family in glmer for accuracy data). 

We would like to thank the Reviewer for this comment which led to a correction of our script. In the new script, the models which include accuracy as the dependent variable have been changed to “glmer” and the specification “family=binomial” has been specified in the code. The old tables have also been replaced by new ones (S1 Table, S2 Table, S9 Table, Supporting Information) derived from the correct models (i.e., “glmer”). Importantly, the significant effects revealed by the corrected models remain the same as the ones derived from the old models.

● The authors write that, for all the analyses, the final model included both participants and test items as random slopes. However, the script online only shows intercepts for the two random effects.

We thank the Reviewer for spotting this error. In the revised version, we replaced the incorrect word “slopes” with “intercepts”.

Pp. 16-17: “The final model included both participants and test items as random intercepts”.

● In addition, the procedure for dealing with convergence issues needs to be clarified. Bates et al. (ref 77) offer minimal suggestions; if the authors used an optimizer, which one did they use? If needed, a variety of papers address this convergence issues (e.g., Barr et al., 2013; Barr et al., 2013, in Frontiers, on corrections; Bates et al., 2015, and Matuschek et al., 2017, on parsimonious models; Brauer & Curtin, 2018). 

We added a reference to Barr et al. (2013) on p. 16. To be more specific, we started by removing the interactions in the slopes, then we proceeded to remove the slopes with lower explained variance until convergence was reached.

● I wonder why the authors ran two sets of analyses, one including monolinguals and the other excluding them. Wouldn’t run only the most inclusive analysis and then set up paired comparisons be acceptable (e.g., with emmeans)? 

RQ II seeks to untangle the influence of sociolinguistic variables within the bilingual experience, so the second set of analyses focuses exclusively on the bilingual and bidialectal groups. Consequently, the monolingual participants were not included in this phase. We did not have relevant data from this group, such as the percentage of language switching, to include in the analyses due to the monolingual nature of this group. 

● If would also simplify the Results section. For example, on p. 17 (with reference to Table 3), I am not sure I understand the need to rerun analyses with a different group as baseline. The authors write that contrasts were set to sum contrasts, which is what appears in the R script. But the results seem to be set up in terms of treatment (or dummy) coding, in which one group (e.g., monolinguals) is the baseline. Please clarify. 

We undertook this step because in R, when sum contrasts are specified, the comparative analyses follow the sequence designated in the code and the outcome does not offer the values of the baseline group. Consequently, if we are working with four distinct language groups, we can only observe the values for the remaining three groups. Hence, it becomes necessary to execute the same model while configuring the sum contrast with different groups as the baseline level to facilitate the examination of the group that was previously selected as baseline. In the revised version of our paper, we have presented the output from the model rerun with different sum contrasts in the Supporting Information.

● What were the theoretical reasons to include Gender and Age as a fixed factor? These factors were not discussed in the Introduction. Do we expect any differences between/among genders? 

As outlined in the model descriptions provided on pp. 16 and 17, we set Gender and Age as control factors rather than fixed factors. This choice led us to not extensively delve into their theoretical implications, while at the same time controlling for unexplained variance in the model. In the revised version, we have included a brief mention of the possible influence of Gender and Age on language processing on p. 9. 

P. 9: “In addition, sociodemographic variables that have been found to potentially impact language processing, such as gender [78] and age [79], will be taken into account as control factors.”

● Accuracy and RT analyses had the same number of data points. Did the RT analysis also include incorrect responses and grammatically correct sentences? Did accuracy data also include responses to grammatically correct sentences? Please clarify what set of data are typically analyzed for this task and which sets were used in the study. 

Since we aim to investigate whether participants with different linguistic backgrounds variably detect Subject-Verb agreement mismatches, our analyses were run on both accuracy and RTs of non-grammatical stimuli (i.e., 40 test items). We clarified this point on p. 15.

P. 15: “Since all 108 participants encountered all test items, which consisted of 40 ungrammatical sentences with Subject-Verb agreement mismatches, 8640 data points were collected, 4320 for each measure (i.e., acceptability judgements and RTs). Data analyses included both accurate and inaccurate responses to the test items”.

● VIF values are not reported, so it is difficult to estimate the extent of collinearity issues. 

We addressed this comment from the Reviewer by adding VIF tables in the Supporting Information (S3 Table, S4 Table, S7 Table, and S8 Table).

● Figures and tables: Accuracy analyses were carried on accurate and inaccurate responses (a combination of values on the Likert acceptability scale). Figure 2 is providing mean accuracies for the various groups; thus, I am not sure what Figure 1 adds. Please clarify or remove the figure.

We followed the Reviewer’s suggestion, and we kept only Figure 2, which has been renumbered as 1 in the new version of the manuscript. 

● Log RT are difficult to interpret; can raw RT be reported somewhere? 

We understand the Reviewer's concern about Log RTs being hard to interpret. Consequently, we re-transformed the predicted values and presented them in milliseconds in the new version of the manuscript. 

● Instead of the included tables in text, I think a more useful (because more direct) way to report the result of the linear mixed models analyses would be to report important values (e.g., t and coefficients) in the text and add the output of the analyses (the models that were tested) as supplementary materials. 

Following the Reviewer’s suggestion, we moved all the tables from the GLME and LME models in the Supporting Information and we included t values within the main body of the text.

● Relatedly, I am confused about “interactions” between fixed factors and dependent variables (e.g., Table 5). Expressions regarding interactions between fixed factors and dependent variables are frequent in the manuscript. I suggest replacing them, as the term “interaction” is typically reserved for fixed factors (i.e., when a fixed factor moderates the effect of another factor on the dependent variable). 

Following this Reviewer’s recommendation, we replaced the term "interaction" with "effect" throughout our manuscript. The only exceptions to this modification are instances where we intend to convey the influence of a fixed factor on another variable, subsequently affecting the dependent variable. We illustrate the relevant changes below:

P. 20: “a statistically significant effect of age is found (t = 3.76)”.

P. 21: “there is a main effect of age and register”.

P. 21: “when examining the main effect of the percentage of use of Italian…”.

P. 21: “Fig 5. Interaction between the percentage of L2 use and language group.”

P. 21: “There is a statistically significant interaction between language group and the percentage of language switching.”

P. 22: “Fig 6. Interaction between the percentage of language switching and Language group.”

P. 22: “a statistically significant effect of age on RTs is found”.

P. 22: “We find an interaction between language group and percentage of language switching.”.

● To make the Results section easier to read, I suggest linking specific analyses to specific research questions and replacing subsections with more informative titles (instead of main analysis and further analysis). Relatedly, the specific research questions should indicate the critical dependent variable. As mentioned above, I am still unclear why some analyses (e.g., correlations) were carried out on overall RT instead of the RT on mismatched sentences. 

We followed the Reviewer’s advice and updated the title for each set of analysis. The revised titles (pp. 17 and 20) now connect the analyses to our research questions. We also identified the critical dependent variables underpinning our analyses on p. 9. Moreover, as we mentioned in a previous reply, in the revised version, we have underscored that all analyses were conducted on acceptability ratings and response times (RTs) pertaining to our 40 test items, all featuring Subject-Verb agreement mismatches (p. 15). Both accurate and inaccurate responses were considered in the analysis.

● Also, the first sentence of the Discussion clarifies that the goal of the study was “to explore how bilingualism influences the linguistic processing of agreement attraction errors.” However, it seems that analyses were run also on correct responses. In some cases, it is unclear what type of RT were analyzed. For example, on p. 25 (line 507 on), in the Discussion, the authors write about “the comparison between monolingual, bilingual, and bidialectal processing of attraction errors…” referring to Fig. 2 and 3. But it is not clear that the RT in Fig. 3 only refers to attraction errors. 

All our test items feature agreement attraction errors. Spotting them by providing accurate or inaccurate answers is a matter tightly connected to the RTs. Therefore, as explained above, our analyses encompass both acceptability ratings and RTs related to our 40 test items, covering the full dataset of the elicited (accurate and inaccurate) responses.

● Discussion: Given the problems outlined above (i.e., uncertainty regarding the specific dependent variables and analyses), I find it difficult to evaluate the Discussion. My suggestion would be to revise and reorganize the discussion after more targeted analyses. 

We hope that the changes made to the Discussion have improved readability. We outline further changes below.

● But I would like to highlight the following. Lower accuracy in Italian-Agrigentino speakers compared to monolinguals is tied to two factors: lack of a rigid demarcation regarding the context in which the two languages are used (with consecutively less need to monitor for external cues and inhibit one of the two languages, p. 27), and the nature of the relationship between the two languages, whose boundaries seem less rigid than for the other groups of bilinguals included in the study (p. 28; I assume these two factors are not completely independent). The latter aspect would create more tolerance in evaluating grammatical deviations (in the first language); the authors write “This dynamic is very likely to affect sociolinguistic contexts where a standard language and a non-standard language co-exist since the last is not defined by linguistic standardization [88-91].” If Agrigentino is not rigidly demarcated, I wonder how to interpret the 54.64% of language switching reported by speakers of Italian and Agrigentino (a percentage that is numerically higher than the one for Italian-Pavese bilinguals, 47.12%). 

The high rates of switching should be interpreted as an additional indicator of the absence of clear linguistic boundaries. When individuals use two languages that exist on a continuum and that are intricately intertwined (i.e., Italian-Agrigentino bidialectals), they are likely to perceive the strong presence of dialectal elements in their everyday speech. Consequently, when asked about the frequency of switching between dialect and standard Italian, they may indeed report frequent switching practices. However, this should not be interpreted as an isolated use of exclusively dialectal or Italian linguistic elements in their respective languages. Instead, it could correspond to a complex fusion of linguistic elements spanning various levels (i.e., lexicon, syntax, etc). The specific dynamics of language switching that characterize standard Italian and Italian dialects have been described by Ramat (1995). She provided examples from the Italian dialects spoken in Pavia and in Sicily, which closely correspond to the varieties examined in our study. We added some discussion on this point on p. 25.

P. 25: “The higher percentages of language switching reported by Italian-Agrigentino bidialectals (Table 1) further support this claim: in those sociolinguistic contexts where Italian and the dialect overlap across different communicative domains, as happens in Agrigento, the frequent language-switching practices lead to a fusion between the codes, something less likely to happen in contexts where the two codes are kept more separated [99].” 

● Also, is there evidence that the presence of a linguistic continuum causes more tolerance in evaluating certain types of grammatical mismatches in the standard language or is it a speculation? 

It is a speculation that is based on previous literature. The tendency to adopt linguistic forms that cannot be ascribed to one specific grammar has been frequently reported in situations where two linguistic varieties co-exist (p. 26: “This is very likely to occur in sociolinguistic contexts where a standard language and a non-standard language co-exist since the last is not defined by linguistic standardization [105-108]”). Given that sociolinguistic settings involving the coexistence of a standard language and a non-standard language tend to be linked to a greater tolerance for linguistic forms that do not conform to the established norms of either system, we interpreted our findings from Italian-Agrigentino bidialectals in this light. 

Minor points

● P. 5: “…the processing of adjacent Subject-Verb agreement was found to positively correlate with perceptual salience and language use, rather than AoA or proficiency.” Please describe the results of the study more in detail. What is, in this context, perceptual salience? And is perceptual salience important here? Also, which language in language use? L1 or L2? Also, “the processing of adjacent Subject-Verb agreement” refers to accuracy or errors? 

We clarified these points on p. 6.

P. 6: “The roles of age of acquisition (AoA) and proficiency were examined by Sagarra and Rodriguez [58], who found that Spanish monolinguals and English-Spanish bilinguals showed similar sensitivity to agreement violations. In particular, the processing patterns of adjacent Subject-Verb agreement in terms of reading times, gaze duration, and accuracy were found to positively correlate with perceptual salience, defined as “the ability of a stimulus to stand out from the rest and to attract attention by virtue of physical characteristics” [58, p. 16], and with L1 and L2 patterns of use, rather than AoA or proficiency.”

● P. 6, end of first paragraph: “The three language groups did not show any difference in attraction error production, but a positive correlation between attraction rates, verbal WM, and inhibitory control was found in all groups.” This description refers to ref. 59 (the correct order of authors is Veenstra, Antoniou, Katsos, & Kissine, based on the original publication). I think the description is imprecise, as lower attraction rates were found in participants with higher verbal WM skills (so, the correlation was negative). In addition, participants who scored higher on inhibition scales (that is, showed higher switch costs and higher interference, indicating lower inhibition control) experienced higher attraction errors. 

We corrected the order and updated the study description (p. 6).

P. 6: “. Regarding Subject-Verb agreement attraction errors, to the best of our knowledge, only Veenstra, Antoniou, Katsos, and Kissine [61] compared bilingual and bidialectal speakers. The tested populations were monolingual Dutch-speaking children, bilingual Dutch- and French-speaking children, and bidialectal Dutch- and West Flemish-speaking children. The three language groups did not show any difference in attraction error production, but a correlation between attraction rates, verbal WM, and inhibitory control was found in all groups: participants with higher WM skills exhibited lower attraction rates compared to participants with weaker inhibitory control, who made more attraction errors.”

● Based on the script, multiple packages were used to analyze the data. However, the packages are not referenced in the References section. It is standard practice to include, in that section, all the R packages used for the analysis, along with the software used for R (e.g., R Studio) and their version, as different package versions might yield different results. Please refer to Meteyards & Davis (2020) for best practices for linear mixed-effects models in psychological science (another good paper is Brown, 2021). 

We included the reference to the most important R packages we used to run our analyses (p. 15).

● A number of concepts are introduced but not defined, for example, register, dilalia, disglossia. Please define these terms when they first appear in the text. 

We added a definition of the mentioned concepts on p. 13 for “Register”, and on p. 27 for “Dilalia” and “Diglossia”. 

P. 13: “Linguistic register is defined as a variety of language shaped by different situational settings [84].”

P. 27: “While in a situation of diglossia, users associate each of their codes to specific social contexts (i.e., “high code”/standard Italian in official and formal settings vs. “low code”/dialect in informal settings), in a dilalic context, the two codes can overlap in different communicative situations [115].”

● Please spell out acronyms before using them (e.g., LSBQ, ISTAT; p. 9 and 27, respectively).

In the revised version, they have been spelled out.

● Grammatical illusions are introduced on p. 3; the first example is on p. 8, in the Method section. I would provide an example at the outset (pp. 3 or 4) for readers not familiar with this literature.

Αn example has been added on p. 4.

P. 4: “(1) *The key to the cabinets are rusty. [28]

● Method: the participant section typically precedes the description of the task (materials and procedure). What was the reason to invert their order? 

We changed the order of the sections following the reviewer’s suggestion.

● Results: Some effects were manipulated and statistical significant, but not much emphasis is put on them (e.g., register). Are they theoretically relevant?

Potentially yes, but as the literature on the impact of register on attraction errors is quite scarce, this is a tentative expectation. As we explained in the previous version of the manuscript, the discussion of the effects related to Register and Animacy will be fully addressed in a separate paper. However, in the revised version we have taken the opportunity to provide a more detailed discussion of these effects on p. 13. In doing so, we have also outlined the rationale behind the inclusion of these effects in our experimental design.

P. 13: “We expect to find an effect of register variation on the detection of Subject-Verb agreement mismatches, further modulated by the participants’ linguistic background. In terms of animacy, we aim to replicate previous findings [28], with animate distractors eliciting stronger attraction effects compared to inanimate distractors.”

● P. 25, first sentence of the discussion:“The goal of the present study is to explore how bilingualism influences the linguistic processing of agreement attraction errors.” I think it would be useful to add something regarding the language in which the errors were measured (Italian, which was the native language but perhaps not the only native language).

In the revised version, we clarified this in the first paragraph of the Discussion section, on p. 22.

P. 22: “The goal of the present study is to explore how bilingualism influences the linguistic processing of agreement attraction errors. The language of testing is Italian, which amounts to the only native language of the monolingual group and one of the native languages of the bilingual/bidialectal groups.”

● P. 26: “it is possible that sentences that looked somewhat incorrect or dubious …”: I think “looked” should be replaced by “sounded,” as the stimuli were presented auditorily.

It has been corrected.

● P. 27, line 541: I think the word “language” follows “standard”.

It has been corrected.

Comments by Reviewer 3

● Summary and evaluation: This work presents a very interesting study that highlights the importance of taking into account the bilinguals’ profile and their socio-linguistic context for the comparison between monolinguals and bilinguals in tasks that involve grammatical illusions/agreement errors. It is also an innovative work, as it introduces the specific comparison between bilinguals of "standard/official" languages and bilinguals who use minority languages or dialects as L2. The two research questions are relevant and refer to these two different levels: one more typical of experimental psycholinguistics and another broader socio-linguistic one. Being able to bring both levels together and find satisfactory answers is, without a doubt, an ambitious goal. It seems that the authors succeed, at least partially. The methodology is adequate, the task is a grammatical acceptability task involving auditory stimuli, and the materials used are well designed and controlled. The analyses are also correct. 

We would like to express our gratitude to the Reviewer for the positive assessment of our work. In the following points, we will describe the changes we applied.

● However, it is somewhat surprising that the manipulation of animacy and linguistic register is not justified and that the results obtained in relation to these factors are not presented here. I would like the authors to comment on the reasons for this decision and, even briefly or in a footnote, to indicate why they have included them and what differences can be expected between the different groups of participants in relation to these variables. 

In the revised version, we have explained our decision to include manipulations of Animacy and Register on p. 13. 

P. 13: “We expect to find an effect of register variation on the detection of Subject-Verb agreement mismatches, further modulated by the participants’ linguistic background. In terms of animacy, we aim to replicate previous findings [28], with animate distractors eliciting stronger attraction effects compared to inanimate distractors.”

● Finally, without questioning the rigor of this study, I would like a final explicit reflection from the authors regarding its scope/limitations. Given the enormous variability across different sociolinguistic contexts, and even across individuals, do the authors estimate that the size of the groups is sufficient and that these results would be easily replicable in the same context or in a different one? One might think that the task of finding consistent effects in cognitive/linguistic processing across bilingual studies is infeasible. Or that we could fall into ad hoc explanations that justify any type of effect, no matter how surprising it may seem. 

In the revised version, we have added a paragraph concerning the limitations of our work, also touching upon potential future avenues of research within the realm of bilingualism, bidialectalism, and language processing. This paragraph has been added in the Conclusion session, on pp. 32-33.

Pp. 32-33: “Among the limitations of our work, we would like to highlight the issue of adequate sampling and representation of minority language users who come from multidialectal communities that show considerable variation. Future work on larger and more diverse samples could add to our claims as well as clarify the impact of individual differences among participants. Furthermore, replicating our research while using different language groups will provide further insights into the role of various sociolinguistic variables, helping us to pinpoint the key factors that affect language processing. All in all, if we manage to ascribe bilingual effects to specific environmental conditions, the apparent inconsistency of results in bilingualism research could possibly be justified and explained as variation caused by the distinct sociolinguistic factors that synthesize every linguistic experience.”

---

## [Decision Letter · Decision Letter 1]

19 Dec 2023

PONE-D-23-23745R1The role of minority language bilingualism in spotting agreement attraction errors: Evidence from Italian varietiesPLOS ONE

Dear Dr. Masullo,

Thank you for submitting your manuscript to PLOS ONE. I am delighted to let you know that after the careful reviews from two out the three previous reviewers who had reviewed the manuscript previously, the manuscript will be accepted for publication in PlosOne if you address the comments of the second reviewer. I agree with her regarding the causal interpretations of regression coefficients since they can only be justified by relying on much stricter assumptions than are needed for predictive inference. Thus, soften please the statements made in this regard. Please submit your revised manuscript by Feb 02 2024 11:59PM. If you will need more time than this to complete your revisions, please reply to this message or contact the journal office at plosone@plos.org. Please include the following items when submitting your revised manuscript:A rebuttal letter that responds to each point raised by the academic editor and reviewer(s). You should upload this letter as a separate file labeled 'Response to Reviewers'.A marked-up copy of your manuscript that highlights changes made to the original version. You should upload this as a separate file labeled 'Revised Manuscript with Track Changes'.An unmarked version of your revised paper without tracked changes. You should upload this as a separate file labeled 'Manuscript'.If applicable, we recommend that you deposit your laboratory protocols in protocols.io to enhance the reproducibility of your results. Protocols.io assigns your protocol its own identifier (DOI) so that it can be cited independently in the future. For instructions see: https://journals.plos.org/plosone/s/submission-guidelines#loc-laboratory-protocols. Additionally, PLOS ONE offers an option for publishing peer-reviewed Lab Protocol articles, which describe protocols hosted on protocols.io. Read more information on sharing protocols at https://plos.org/protocols?utm_medium=editorial-email&utm_source=authorletters&utm_campaign=protocols.

We look forward to receiving your revised manuscript.

Kind regards,

Montserrat Comesaña Vila

Academic Editor

PLOS ONE

Journal Requirements:

Reviewers' comments:

Reviewer's Responses to Questions

**Comments to the Author**

1. If the authors have adequately addressed your comments raised in a previous round of review and you feel that this manuscript is now acceptable for publication, you may indicate that here to bypass the “Comments to the Author” section, enter your conflict of interest statement in the “Confidential to Editor” section, and submit your "Accept" recommendation.

Reviewer #2: (No Response)

Reviewer #3: All comments have been addressed

2. Is the manuscript technically sound, and do the data support the conclusions?

Reviewer #2: (No Response)

Reviewer #3: Yes

3. Has the statistical analysis been performed appropriately and rigorously? 

Reviewer #2: (No Response)

Reviewer #3: Yes

4. Have the authors made all data underlying the findings in their manuscript fully available?

Reviewer #2: (No Response)

Reviewer #3: Yes

5. Is the manuscript presented in an intelligible fashion and written in standard English?

Reviewer #2: (No Response)

Reviewer #3: Yes

6. Review Comments to the Author

Reviewer #2: I appreciate the authors’ care in addressing the many comments that I submitted. While I think that the manuscript has improved substantially, I think some aspects of the study remain to be clarified. They are outlined below.

- Participants

o In their revision and response, the authors did not clarify why 152/278 participants did not complete the task. I think this information should be included in the manuscript.

- Stimuli

o Did the authors create the stimuli? Or where borrowed from previously published work? Please clarify and cite pertinent work, if needed.

- Variables

o In their response, the authors wrote that the DV concerned their performance to the critical 40 ungrammatical sentences, in terms of accuracy and RT. They also wrote “Data analyses included both accurate and inaccurate responses to the test items” (p. 15, lines 341-342) and “Moreover, we examined whether the acceptability rates influence the RTs” (p. 17, line 382). I am not sure how acceptability rates can influence RT, as we are discussing two dependent variables and a relationship about covariation. Also, it is still unclear to me why RT analyses were calculated for both accurate and inaccurate responses. Given that the main hypotheses in the study concern attraction errors, the analyses could have focused on the RT of accurate responses (answers 1 and 2 to sentences with attraction errors). Why including RT for answers 3, 4, 5? What is the justification, theoretically speaking? Much of the Results and Discussion sections concerns differences in RT between groups, so I think it is very important to explain this choice.

- Analyses

o P. 16: Information about how the authors dealt with convergence issues should be added to the paper. (“To be more specific, we started by removing the interactions in the slopes, then we proceeded to remove the slopes with lower explained variance until convergence was reached.”)

o Pp. 21-22: RTs analyses. I assume that both accurate and inaccurate trials were included? [please see my comment in the “Variables” section.]

- Figures

o In general: all figures should include information on how variability in the data is represented (e.g., SD, SE). They only include the variables is on the X and Y axes.

o P. 19, Figure 3. The legend states the following: “Fig 3. Distribution of RTs for each language group. The x-axis represents how the distribution of judgements aligns with RTs across various language groups. The y-axis reports RTs in milliseconds.” I am not sure I understand what alignment means here, as the various types of judgments are not clearly distinguishable.

o P. 20, Figure 4. I commented on the relationship between these two dependent variables in the Variables section. Regardless, I wonder if instead of “Fig 4. Interaction between RTs and Judgement for each language group” are more apt title would be ““Fig 4. Relationship between RTs and Judgement for each language group.”

o P. 22, Figure 5. “Fig 5. Interaction between the percentage of L2 use and language group. The x-axis shows the percentage of time using the L2, while the y-axis reports RTs in milliseconds.” There is only one factor represented in the figure (percentage of language used).

- Discussion

o P. 29: “In particular, we find a positive correlation between RTs and the percentage of L2 use.” The correlation is negative, based on Fig. 5: the higher the L2 use, the shorter the RT. Also, based on previous analyses, I assume that RTs were calculated on all five types of responses, thus including inaccurate responses. If this is incorrect (so, only accurate responses were included), please disregard this comment. If it is correct, again, I wonder what the inclusion of both correct and incorrect answers tells us, in terms of the hypotheses assessed here.

o I am unclear about the last paragraph on p. 29. The discussion concerns the data in Fig. 6, with two different patterns of RTs in Pavese (positive correlation with percentage of switching) and standard bilinguals (negative correlation). I am not sure what the following sentence refers to: “However, the absence of significant correlations between RTs and accuracy does not allow further speculations on the possible advantages of language switching for our bilingual participants.” Please clarify.

Finally, in their response, the authors wrote:

“We apologise for the confusion created by the use of inaccurate terminology in the previous version of the manuscript. This is not a correlation study. For the analyses, we used regression models, which reveal whether and in what way the independent variables (i.e., fixed effects) have an impact on the dependent ones (accuracy of the judgments and response time). Differently from correlation analyses, which assess the strength of the relationship between two variables, regression analyses uncover causal relationships between the variables, enabling us to discern the effects they exert on one another. Hence our choice to use terms such as “effect” and “impact”.”

Perhaps I am adopting a conservative approach, but if there is no manipulation (here, for example, language experience, gender, age are all subject variables), we cannot speak of causality. Causality is inferred based on the design of the study, not the statistical analyses used. So, I advise caution in both describing statistical effects and drawing conclusions when causality is drawn from a study with a non-experimental design.

Reviewer #3: (No Response)

7. PLOS authors have the option to publish the peer review history of their article (what does this mean?). If published, this will include your full peer review and any attached files.

Reviewer #2: No

Reviewer #3: No

---

## [Author Response · Author response to Decision Letter 1]

15 Jan 2024

Response letter

We would like to thank the Editor, Prof. Montserrat Comesaña Vila, and all the Reviewers for their positive feedback on our revised work. In this letter, we address all the points raised by the second Reviewer. The changes made to address the Reviewer’s observations appear in red in the revised version of the manuscript and are also reported in this response letter.

Comments by Reviewer 2

I appreciate the authors’ care in addressing the many comments that I submitted. While I think that the manuscript has improved substantially, I think some aspects of the study remain to be clarified. They are outlined below.

• Participants

In their revision and response, the authors did not clarify why 152/278 participants did not complete the task. I think this information should be included in the manuscript. 

We added the required clarification on page 10 of the revised version of the manuscript.

P. 10: “In particular, the 152 participants who did not complete the task were excluded because they only filled one section of the experiment, namely the background questionnaire, without starting or, in some cases, completing the acceptability judgement task before the end of the data collection session.”

• Stimuli

Did the authors create the stimuli? Or where borrowed from previously published work? Please clarify and cite pertinent work, if needed. 

This piece of information has been added to the revised version of the manuscript, on page 12. 

P. 12: “The stimuli were specifically created for this study and constitute original material available at: https://osf.io/j4zqg/?view_only=e52f1e4facb9474984148cefac087b51.”

• Variables

o In their response, the authors wrote that the DV concerned their performance to the critical 40 ungrammatical sentences, in terms of accuracy and RT. They also wrote “Data analyses included both accurate and inaccurate responses to the test items” (p. 15, lines 341-342) and “Moreover, we examined whether the acceptability rates influence the RTs” (p. 17, line 382). I am not sure how acceptability rates can influence RT, as we are discussing two dependent variables and a relationship about covariation. 

Although acceptability rates constitute the dependent variable of one set of our analyses (“Accuracy analyses”, pp. 16, 18, 21), they are not set as dependent variables in all the models included in our analyses. In the analyses of RTs, where RTs were set as the dependent variable (“RT analyses”, pp. 17, 19, 22), we decided to include acceptability rates as a fixed factor, as specified on page 17 (“As fixed effects, we included…the acceptability judgement given to the stimulus (scaled)”), to observe any potential relation between distinct judgements on a 5-point Likert scale and the corresponding RTs. Thus, in the case of RT analyses, acceptability rates are not the dependent variable anymore. They are instead treated as a potential factor modulating RTs. To clarify this point, we changed the phrasing of the sentence cited by the Reviewer as follows:

P. 17: “Moreover, we examined whether the acceptability rates influence the RTs were modulated by acceptability judgements.”

o Also, it is still unclear to me why RT analyses were calculated for both accurate and inaccurate responses. Given that the main hypotheses in the study concern attraction errors, the analyses could have focused on the RT of accurate responses (answers 1 and 2 to sentences with attraction errors). Why including RT for answers 3, 4, 5? What is the justification, theoretically speaking? Much of the Results and Discussion sections concerns differences in RT between groups, so I think it is very important to explain this choice.

We have added a paragraph on page 15 of the revised manuscript, including two new references [86, 87], to address this comment from the Reviewer.

P. 15: “The inclusion of both accurate and inaccurate responses and their corresponding RTs was done to comprehensively observe participants’ behavior regarding agreement attraction errors, which is the main purpose of this study. By including RTs of both accurate and inaccurate responses, we seek to highlight potential trends in the time needed to give (in)correct responses, which have been highlighted in previous literature [60, 86]. In particular, recent research on the processing of Subject-Verb agreement mismatches showed that inaccurate judgements are associated with slower RTs compared to accurate judgements [87]. Furthermore, given that Italian-Pavese and Italian-Agrigentino bidialectals have never been examined in the processing of Subject-Verb agreement mismatches, and in general, in language processing research, we opted not to exclude a priori a significant portion of our database to entirely observe the processing behavior of these unstudied populations”.

• Analyses

o P. 16: Information about how the authors dealt with convergence issues should be added to the paper. (“To be more specific, we started by removing the interactions in the slopes, then we proceeded to remove the slopes with lower explained variance until convergence was reached.”)

We added this piece of information in the revised version of the manuscript.

P. 16: “To be more specific, we started by removing the interactions in the slopes, then we proceeded to remove the slopes with lower explained variance until convergence was reached”.

P. 17: “Again, we started by removing the interactions in the slopes, then we removed the slopes with lower explained variance until reaching convergence”.

o Pp. 21-22: RTs analyses. I assume that both accurate and inaccurate trials were included? [please see my comment in the “Variables” section.]

Yes, both accurate and inaccurate trials were included. As we specified in the answer to the Reviewer’s comment in the “Variables” section of the present Response Letter, we have added a paragraph on page 15 of the revised version of the manuscript. In this new paragraph, we explain our choice of including RTs from both accurate and inaccurate acceptability judgements.

• Figures

o All figures should include information on how variability in the data is represented (e.g., SD, SE). They only include the variables is on the X and Y axes.

The required pieces of information have been added to all captions of the revised version of the manuscript.

P. 18: “Fig 1. Accuracy rates for each language group. The x-axis represents the language groups, while the y-axis shows the mean accuracy rates from 0 (i.e., “Inaccurate”) to 1 (i.e., “Accurate”). The vertical lines represent standard errors”.

P. 20: “Fig 2. RTs for each language group. The x-axis represents the language groups, while the y-axis shows RTs in milliseconds recorded for each language group. The vertical lines represent standard errors”.

P. 20: “Fig 3. Distribution of RTs for each language group. The x-axis represents how the distribution of judgements aligns with RTs across various language groups. The y-axis reports RTs in milliseconds. The violin shapes represent data density, while the box plots represent standard deviations”.

P. 21: “Fig 4. Interaction between RTs and Judgement for each language group. Interaction between Acceptability Judgment and Language group on RTs. The x-axis shows the acceptability judgements given to the stimuli. The y-axis reports RTs in milliseconds. The error ribbons represent 95% confidence intervals”.

P. 23: “Fig 5. Main effect of the percentage of L2 use on RTs Interaction between the percentage of L2 use and language group. The x-axis shows the percentage of time using the L2, while the y-axis reports RTs in milliseconds. The error ribbon represents 95% confidence interval”.

P. 23: “Fig 6. Interaction between the percentage of language switching and Language group. The x-axis shows the percentage of language switching. The y-axis reports RTs in milliseconds. The error ribbon represents 95% confidence interval”.

o P. 19, Figure 3. The legend states the following: “Fig 3. Distribution of RTs for each language group. The x-axis represents how the distribution of judgements aligns with RTs across various language groups. The y-axis reports RTs in milliseconds.” I am not sure I understand what alignment means here, as the various types of judgments are not clearly distinguishable. 

Thanks to this Reviewer’s comment, we modified the caption of Figure 3 as follows:

P. 20: “Fig 3. Distribution of RTs for each language group. The x-axis represents how the distribution of judgements aligns with RTs across various language groups. The y-axis reports RTs in milliseconds. The violin shapes represent data density, while the box plots represent standard deviations.”.

o P. 20, Figure 4. I commented on the relationship between these two dependent variables in the Variables section. Regardless, I wonder if instead of “Fig 4. Interaction between RTs and Judgement for each language group” are more apt title would be ““Fig 4. Relationship between RTs and Judgement for each language group.”

We appreciate the Reviewer’s suggestion. Still, we believe it is important to remark that the figure represents the significant interaction between the acceptability judgments and language groups found in the statistical model of RTs. We modified the caption as follows:

P. 21: “Fig 4. Interaction between RTs and Judgement for each language group. Interaction between Acceptability Judgment and Language group on RTs. The x-axis shows the acceptability judgements given to the stimuli. The y-axis reports RTs in milliseconds. The error ribbons represent 95% confidence intervals.

o P. 22, Figure 5. “Fig 5. Interaction between the percentage of L2 use and language group. The x-axis shows the percentage of time using the L2, while the y-axis reports RTs in milliseconds.” There is only one factor represented in the figure (percentage of language used).

We changed the caption of Figure 5 as follows:

P. 23: “Fig 5. Main effect of the percentage of L2 use on RTs Interaction between the percentage of L2 use and language group. The x-axis shows the percentage of time using the L2, while the y-axis reports RTs in milliseconds. The error ribbon represents 95% confidence interval.”.

• Discussion

o P. 29: “In particular, we find a positive correlation between RTs and the percentage of L2 use.” The correlation is negative, based on Fig. 5: the higher the L2 use, the shorter the RT. Also, based on previous analyses, I assume that RTs were calculated on all five types of responses, thus including inaccurate responses. If this is incorrect (so, only accurate responses were included), please disregard this comment. If it is correct, again, I wonder what the inclusion of both correct and incorrect answers tells us, in terms of the hypotheses assessed here.

Thanks to the Reviewer’s comment, we substituted the adjective “positive” with “negative” in the revised version of the manuscript.

P. 30: “In particular, we find a negative positive correlation relation between RTs and the percentage of L2 use.”.

As commented in earlier sections of this response letter, we opted to incorporate RTs for both accurate and inaccurate responses to ensure the inclusion of significant data portions in our analyses. Since Italian-Pavese and Italian-Agrigentino bidialectals have never been tested on an AJT involving Subject-Verb agreement mismatches, we wanted to observe behavioral results in their completeness, without precluding insights coming from incorrect responses. Furthermore, if we had excluded incorrect judgements (i.e., 3, 4, and 5), it would have been impossible to highlight the trend illustrated in Figure 4 (i.e., longer RTs for incorrect judgements).

In the specific case of the relation between RTs and L2 use, we considered the possibility of a clearer distinction between languages among participants who use their L2 more frequently. This distinction could result in reduced processing costs, attested by faster RTs, for evaluating a sentence presented exclusively in one of their linguistic systems, regardless of the accuracy of the judgment. Including both correct and incorrect responses allowed us to observe the general RT tendencies of different bilingual and bidialectal groups concerning specific language use variables.

o I am unclear about the last paragraph on p. 29. The discussion concerns the data in Fig. 6, with two different patterns of RTs in Pavese (positive correlation with percentage of switching) and standard bilinguals (negative correlation). I am not sure what the following sentence refers to: “However, the absence of significant correlations between RTs and accuracy does not allow further speculations on the possible advantages of language switching for our bilingual participants.” Please clarify.

The sentence has been modified in the revised version of the manuscript. 

Pp. 30-31: “However, the absence of a main effect of switching and the lack of interaction between switching, the opposite pattern found in Italian-Spanish bilinguals and Italian-Pavese bidialectals, and language group in the further analysis of bilinguals’ and bidialectals’ accuracy rates does not allow further speculations on the possible advantages of language switching for our bilingual participants.”

o Finally, in their response, the authors wrote:“We apologise for the confusion created by the use of inaccurate terminology in the previous version of the manuscript. This is not a correlation study. For the analyses, we used regression models, which reveal whether and in what way the independent variables (i.e., fixed effects) have an impact on the dependent ones (accuracy of the judgments and response time). Differently from correlation analyses, which assess the strength of the relationship between two variables, regression analyses uncover causal relationships between the variables, enabling us to discern the effects they exert on one another. Hence our choice to use terms such as “effect” and “impact”. Perhaps I am adopting a conservative approach, but if there is no manipulation (here, for example, language experience, gender, age are all subject variables), we cannot speak of causality. Causality is inferred based on the design of the study, not the statistical analyses used. So, I advise caution in both describing statistical effects and drawing conclusions when causality is drawn from a study with a non-experimental design.

We thank the Reviewer for giving us the possibility of clarifying this point. As the Reviewer correctly pointed out, we should make clear that the relations between variables are not necessarily causal. In this light, we modified some parts of the Result and Discussion sections, to soften any statement about the interaction between variables that could have been interpreted as a strong causal relation. In this way, we delineate that the variation of specific factors within the bilingual experience can coincide with changes in accuracy/response times (RTs), without necessarily suggesting causal relationships.

Pp. 20-21: “Interestingly, there is an interaction between acceptability judgements and language groups (t = -2.25, S5 Table, Supporting Information), revealing that while a significant difference between the Italian-Agrigentino bidialectal group is slower and than the monolingual group (t = -2.25, S5 Table, Supporting Information). As Fig 4 shows, monolingual participants show a prominent difference between RTs associated with accurate vs. inaccurate judgements, such that inaccurate judgements are associated with longer RTs, the difference is less pronounced for Italian-Agrigentino bidialectal speakers.”

P. 22: “Our second GLME with accuracy as the dependent variable was run to determine whether the percentage of use of Italian vs. the other majority or minority language, together with the frequency of language switching, modulates has an impact on accuracy rates.”

P. 22: “we are interested in seeing whether variables related to the language practices of bilingual participants with minority vs. majority languages have an impact on modulate RTs.”

P. 30: “our findings demonstrate that language processing outcomes can be attributed to significantly change together with variables associated with specific language practices.”

P. 30: “In particular, we find a negative positive correlation relation between RTs and the percentage of L2 use.”

P. 30: “Besides the percentage of L2 use, another sociolinguistic factor that seems to plays a role on RTs is language switching.”

P. 30: “The positive correlation negative relation between switching and response latencies may suggest support the hypothesis that the constant juggling between two languages trains the parser.”

---

## [Editor Report · Decision Letter 2]

30 Jan 2024

The role of minority language bilingualism in spotting agreement attraction errors: Evidence from Italian varieties

PONE-D-23-23745R2

Dear Dr. Camilla Masullo,

We’re pleased to inform you that your manuscript has been judged scientifically suitable for publication and will be formally accepted for publication once it meets all outstanding technical requirements.

Kind regards,

Montserrat Comesaña

Academic Editor

PLOS ONE

---

## [Editor Report · Acceptance letter]

17 Feb 2024

PONE-D-23-23745R2 

PLOS ONE

Dear Dr. Masullo, 

I'm pleased to inform you that your manuscript has been deemed suitable for publication in PLOS ONE. Congratulations! Your manuscript is now being handed over to our production team.

Kind regards, 

on behalf of

Dr. Montserrat Comesaña Vila 

Academic Editor

PLOS ONE